

# Automated Wind Turbine Wake Characterization in Complex Terrain

Rebecca J. Barthelmie[1] and Sara C. Pryor[2]

[1]Sibley School of Mechanical and Aerospace Engineering, Cornell University, Ithaca, New York
[2]Department of Earth and Atmospheric Sciences, Cornell University, Ithaca, New York

*Correspondence to*: Rebecca J. Barthelmie (rb737@cornell.edu)

**Abstract**

An automated wind turbine wake characterization algorithm has been developed and applied to a dataset of over 19,000 scans
measured by scanning Doppler lidar at Perdigão over the period January to June 2017. The algorithm correctly identifies the wake
centre position in 62% of possible wake cases, 46% having a clear and well-defined wake centre while 16% have split centres or
multiple lobes. Only 5% of cases are not detected, the remaining 33% could not be categorized either by the algorithm or
subjectively, mainly due to the complexity of the background flow. Average wake centre heights categorized by inflow wind
speeds are shown to be initially lofted (to 2 rotor diameters, $D$) except when the inflow wind speeds exceed 12 ms$^{-1}$. Even under
low wind speeds, by 3.5 $D$ downstream of the wind turbine, the mean wake centre position is below the initial wind turbine hub-
height and descends broadly following the terrain slope. However, this behaviour is strongly linked to hour of the day and
atmospheric stability. Overnight and in stable conditions the average height of the wake centre is 10 m higher than in unstable
conditions at 2 $D$ and 17 m higher at 4.5 $D$ downstream of the wind turbine. **Keywords:** Wind turbine; Wakes; Lidar;
Measurements.

## 1   Introduction

### 1.1   Motivation and objectives

Temporal and spatial inhomogeneity of the flow in complex terrain (Kaimal and Finnigan, 1994) increases uncertainty in modelling
and measurements of wind speed, turbulence intensity etc., for wind resource assessment and turbine operating conditions (Sanz
Rodrigo et al., 2017). It also has implications for wind turbine wake generation and propagation (i.e. formation and recovery of the
volume of disturbed air that passes through the wind turbine rotor) (Barthelmie et al., 2013). Most previous research on the
characterization of wind turbine wakes (their meandering, merging and ultimate recovery) and evaluation of wind farm wake
models and wind farm layout optimization to reduce the reduction of power and enhancement of loads on turbines operating within
wind turbine wakes has focussed on applications in relatively flat terrain/offshore (e.g. (Barthelmie and Pryor, 2013)). However,
as wind energy penetration increases, there is a need to develop improved methods for quantifying flow in complex terrain and to
develop methods to optimize layouts to minimize power losses and fatigue loading from turbine-turbine interactions and wind farm
wakes (Barthelmie et al., 2018;Politis et al., 2012). The advection, characteristics and ultimately the dissipation of wakes from
wind turbines located in these environments is strongly dictated by flow features that are, in turn, a product of the terrain length
scales and height (Kaimal and Finnigan, 1994), the presence/absence of vegetation (Finnigan and Belcher, 2004) and/or other
surface heterogeneity/discontinuities (Durran, 1990). The objectives of this paper are to (i) describe observational challenges to
detection and characterization of wind turbine wakes in complex terrain, (ii) present an automated methodology for wake centreline
detection and tracking in data from scanning pulsed Doppler lidar operated in complex terrain, (iii) evaluation of the automated



detection when applied to data from a field experiment conducted in complex terrain and (iv) use the objective detection of wake

centreline location to examine wake behaviour from a single wind turbine as a function of the prevailing atmospheric conditions.

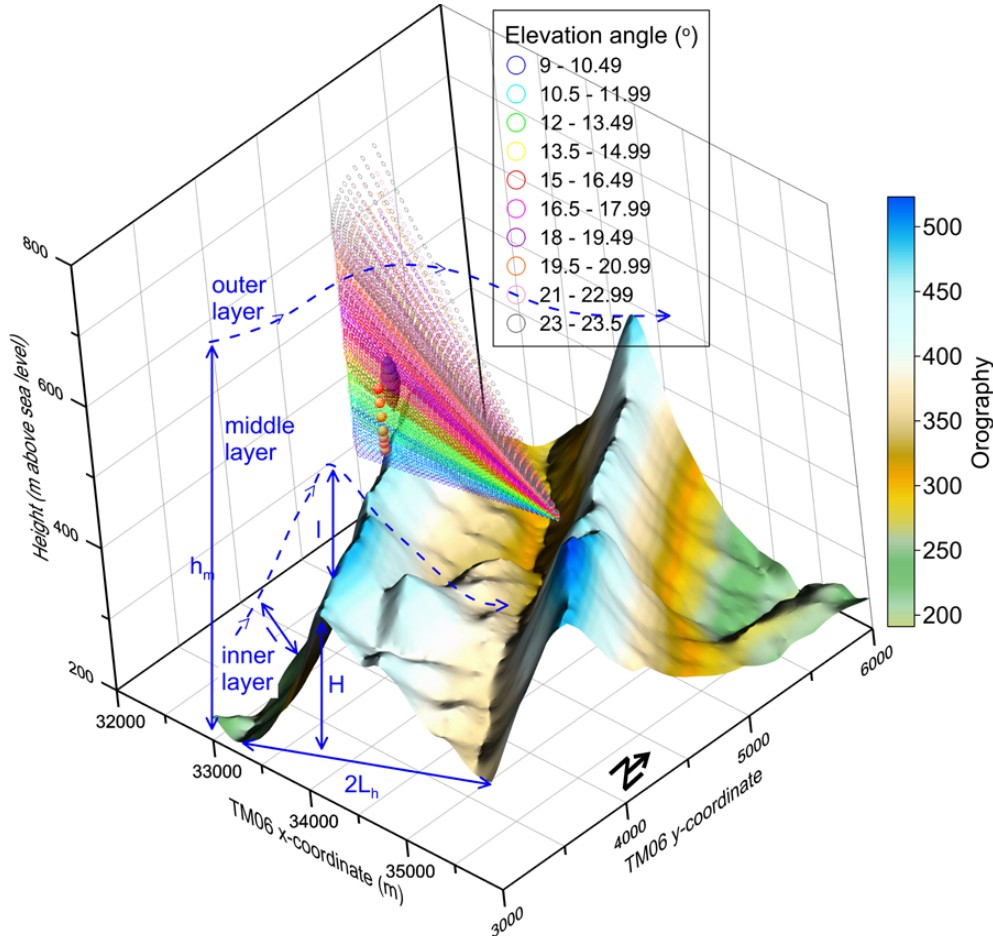

**Figure 1: The orography of the Perdigão study area (shown in Portuguese TM06 coordinates with major spacing at 1 km). The scan**
**points of the arc scans from the scanning Doppler lidar located in the valley are shown for each 30 m range gate for 10 elevation angles**
**from 9-23° and 23 azimuth angles from 193.5-253°. As shown the topography is dominated by double-ridges orientated northwest-**
**southeast. The turbine location on the SW ridge is indicated by a black disc representing the rotor plane and the measurement heights**
**at meteorological mast Tower 20 by the red spheres. The blue lines denote the height ($H=300\ m$) and length scales ($L_h=800\ m$) of the**
**topography for southwesterly flow and the inner layer height $l$ and middle layer height $h_m$.**

## 1.2     Flow in complex terrain

Flow over a hill is characterized by compression, acceleration and lifting of streamlines at the crest (Kaimal and Finnigan, 1994).

In stable conditions, flow may be divided approaching the crest. If the slope is steep enough, a separation bubble forms after the

crest whose depth is of the order of the hill height (Kaimal and Finnigan, 1994). Regardless of the formation of a separation bubble,

a hill wake with marked velocity deficit and enhanced turbulence extends for many hill heights downwind (Kaimal and Finnigan,

1994). The near-surface flow downhill is extremely complex with modifications to the inner and outer layers, even assuming these

continue to exist, and for potential interaction with recirculation zones. For example, flow decelerates downwind of the crest and

the flow becomes detached and a separation bubble forms if the slope is steeper than ~18° and for smaller angles if the slope has

high roughness length (Kaimal and Finnigan, 1994). Flow in the lee of the hill/ridge tends to be more terrain-following in stable





conditions (Hunt et al., 1988) while in neutral and unstable conditions there is a recirculating vortex/separation bubble behind the hill (Ohba et al., 2002). If a separation bubble exists in stable conditions, the reattachment length is similar to that in near-neutral conditions, while the reattachment length is unstable conditions is shorter (Pieterse and Harms, 2013). The scales and dynamical properties of topographically induced flow components and their interaction of wake(s) induced by wind turbines with that flow is

thus a function of the terrain dimensions (ratio of hill height ($H$) to length scale ($L_h$, distance from hill crest to the half-height), prevailing stability, land surface characteristics and vegetation (Figure 1). Here we focus on flow in the m to km range at heights relevant to wind turbines (i.e. up to ~ 150 m a.g.l.) accepting that flow at these scales is also impacted by phenomena at larger scales such as gravity waves and thermally-generated upslope and downslope winds (Durran, 1990).

For near-neutral stability and low hills ,where hill height ($H$) is much smaller than distance from hill crest to the half-height ($L_h$),

the inner layer which is closest to the surface has height, $l$ (Jackson and Hunt, 1975):

$$\frac{\left(l \ln \frac{l}{z_0}\right)}{L_h} = 2\kappa^2 \tag{1}$$

where $z_0$ is the surface roughness length and $\kappa$ is the van Karman constant. This inner layer is typically assumed not to extend beyond the constant stress (surface) layer and to have a height, $l < 0.2\delta$ where $\delta$ is the boundary-layer height. The inner layer height increases as $z_0$ increases but for a range of likely roughness lengths; $z_0 = 0.01$ m to 0.5 m and $L_h = 800$ m, $l$ is ~32 to 60 m. Assuming a logarithmic wind profile (and thus neutral stability), the middle layer height, $h_m$ (i.e. lowest height of the outer layer, see Figure

1), can be defined using (Coppin et al., 1994):

$$h_m \ln^{1/2}(L_h/z_0) = L_h \tag{2}$$

where typically (Coppin et al., 1994):

$$l/h_m \leq 0.1 \tag{3}$$

For $L_h = 800$ m and $z_0$ values given above, $h_m$ is in the range 238-295 m.

These approximations can be extended to include the influence of stability. While noting the limitations on use of similarity theory in complex terrain, using the Monin-Obukhov length $L$ to describe local stability conditions (Kaimal and Finnigan, 1994):

$$L = \frac{\overline{\theta_v} u_*^3}{k g \overline{w'\theta_v'}} \tag{4}$$

where $\theta_v$ is the virtual potential temperature, $u_*$ is the friction velocity, $g$ is acceleration due to gravity and $w'\theta_v'$ is the sensible heat flux (Stull, 1988). The height of the inner layer is then given by (Coppin et al., 1994):

$$\frac{l}{L_h} = 2\kappa^2 \left[\ln\left(\frac{l}{z_0}\right) + \psi_m\left(\frac{1}{L}\right)\right]^{-1} \tag{5}$$

For $z_0 = 0.3$ m, $l$ varies from about 50 m in near-neutral conditions to 42 m in stable conditions increasing to about 63 m in unstable conditions (Table 1).

The top of the middle-layer, $h_m$, in stable conditions (Kaimal and Finnigan, 1994) is given by:

$$h_m \sim L_h \left[\ln\left(\frac{h_m}{z_0}\right) + 5\left(\frac{h_m}{L}\right)\right]^{-1/2} \tag{6}$$

and in unstable conditions (Coppin et al., 1994) by:

$$h_m \sim L_h \left[\frac{\left(1 - 15\frac{h_m}{L}\right)^{-1/4}}{\left(\ln\left(\frac{h_m}{z_0}\right) + \psi_m\left(\frac{h_m}{L}\right)\right)}\right]^{1/2} \tag{7}$$



These approximations imply that for a wind turbine on a low hill, with lowest tip ~ 40 m and highest tip ~ 120 m, the flow impacting the turbine rotor may be inside, close to or above the inner layer depending on the terrain height/length scale/roughness and also extends into the middle layer. For reasonable conditions (e.g. $z_0 = 0.01$ m to 0.5 m, $L_h = 800$ m), that prevail in many currently operational land-based wind farms, the inner layer will intersect the rotor plane, thus this layer has importance for both the inflow

to wind turbines and their wakes. The middle layer will have greatest relevance for the propagation and dispersion of wind turbine wakes at least in the near-wake (Table 1). Conversely, the outer-layer (i.e. flow at heights above $h_m$, and thus the large-scale flow) is generally of lesser importance to wind turbine wake generation, propagation and characteristics, depending on the terrain characteristics.

Consistent with the above, siting wind turbines in complex terrain usually involves placement of the turbines close to/at the top of

ridges where there is maximum speed-up and hence maximum topographic enhancement of the wind resource (Barthelmie et al., 2016a). For homogeneous flow oriented perpendicular to a ridge (Barthelmie et al., 2016a) or at shallow yaw angles from the perpendicular (Barthelmie and Pryor, 2018), this speed-up $\Delta S$ (defined as the location wind speed minus the freestream wind speed normalized by the freestream) is quantified as:

$$\Delta S_{(x,z)} = \frac{U_{(x,z)} - U_{(0,z)}}{U_{(0,z)}} \tag{8}$$

It has its maximum close to the ridge top at heights relatively close to the surface, decreases with height and is larger in conditions

with stable stratification (Emeis et al., 1995). Assuming $\Delta S$ is maximum at a height of $l/3$ (i.e. one-third of the inner layer height, see below) (Kaimal and Finnigan, 1994) under near-neutral stability maximum $\Delta S$ ($\Delta S_{UMax}$) will typically be observed between 14-21 m, although it is worth noting that under different stability conditions:

$$\Delta S_{UMax} = \frac{H}{L_h} \left[ \frac{\overline{U}_{0(h_m)}^2}{\overline{U}_{0(l)} \overline{U}_{0(l/3)}} \right] [1 + \frac{1.8}{\ln(\frac{l}{z_0})}] \tag{9}$$

or for different height/length scales, the speed-up may also impact the flow in the middle layer ($h_m > z > l$) where $\varepsilon$ is approximately equal to $1$ depending on the hill shape (Coppin et al., 1994):

$$\Delta S_{(x,z)} = \frac{H}{L_h} \left[ \frac{\overline{U}_0^2(h_m)}{\overline{U}_0^2(z)} \right] \varepsilon \tag{10}$$

As illustrated by this introduction, placement of wind turbines in complex terrain represents a substantial challenge for efforts to both characterize wind turbine wakes and thus turbine-turbine interactions due to the complexity of the flow field in which they are embedded and to developing robust tools to optimize the layout of large arrays in inhomogeneous environments (Politis et al., 2012).

The topography in the location of the experiment considered herein is dominated by two parallel ridges of almost equal height

(Letson et al., 2018;Fernando et al., 2018) (Figure 1). Flow in this environment is further complicated by the presence of anabatic/katabatic and upslope/downslope flow patterns that result from; i) thermal forcing where within valley winds are defined by local pressure gradients and flow can be detached from winds above, especially under conditions of weak larger scale forcing, ii) strong downward momentum transfer from either vertical turbulent transport or gravity waves i.e. this can occur in either unstable or stable conditions, iii) wind in geostrophic balance being channelled by the valley producing an along valley wind

component and iv) pressure-gradient channelling producing winds along the valley (Whiteman and Doran, 1993).




### 1.3 Characterizing wind turbine wakes

Virtually all wind turbines are installed in wind farms (i.e. multi-turbine arrays) where the interaction of the lower velocity air directly downstream from a wind turbine (wake) on its nearest neighbour reduces the wind speed, and hence the power output, and increases the fatigue loading by increasing the turbulence intensity. Average power losses from large wind turbine arrays due to

wind turbine wakes are reported to be 2-4% on land (El-Asha et al., 2017) and can be 10-20% offshore (Barthelmie et al., 2007). At offshore sites the principal determinant of wake intensity is the freestream wind speed which determines the thrust coefficient of the wind turbine with large relative velocity deficits between cut-in and rated wind speeds (i.e. from ~3-12 ms$^{-1}$) responding to high thrust coefficients (Barthelmie et al., 2013). As wind speeds increase, thrust coefficients decrease and the relative magnitude of the wake velocity decreases. Secondary drivers that also determine power losses due to wind turbine wakes include; wind

direction (that determines spacing between turbines and can be associated with particular atmospheric stability climates or flow regimes (Barthelmie, 1998)), turbulence intensity and atmospheric stability (Barthelmie et al., 2013).

In flat terrain under near-neutral stability, the wake expansion rate (and thus the volume of air with reduced momentum) is assumed to follow (Jensen, 1983):

$$D_w = D_0 + 2kX \qquad (11)$$

where $D_w$ is wake width in rotor diameters ($D$), $D_0$ is the initial wake width, $k$ is the rate of expansion (0.075 is used in WAsP for

land sites (Katic et al., 1986)), which is determined by the factors listed above such as ambient turbulence intensity and $X$ is the distance downstream. Assuming for most wind turbines $D$ ~ hub-height ($WTHH$), the wake expands to $2D$ and hence will impact the ground at approximately $6.6 D$ downstream of the wind turbine (Figure 2). As the wake expands and higher momentum air is drawn into the wake the velocity deficit decreases such that practically, it is difficult to determine the presence and characteristics of wakes from individual wind turbines beyond about $10 D$ downwind, except in stable stratification and/or low surface roughness

conditions when the expansion and meander are more limited. Large wind farm (multiple) wakes can be detected over longer distances (Pryor et al., 2018), particularly in offshore conditions where the generally smooth surface and low turbulence intensity leads to both the deep array effect within arrays and highly persistent whole array wakes (Christiansen and Hasager, 2005;Barthelmie and Jensen, 2010).

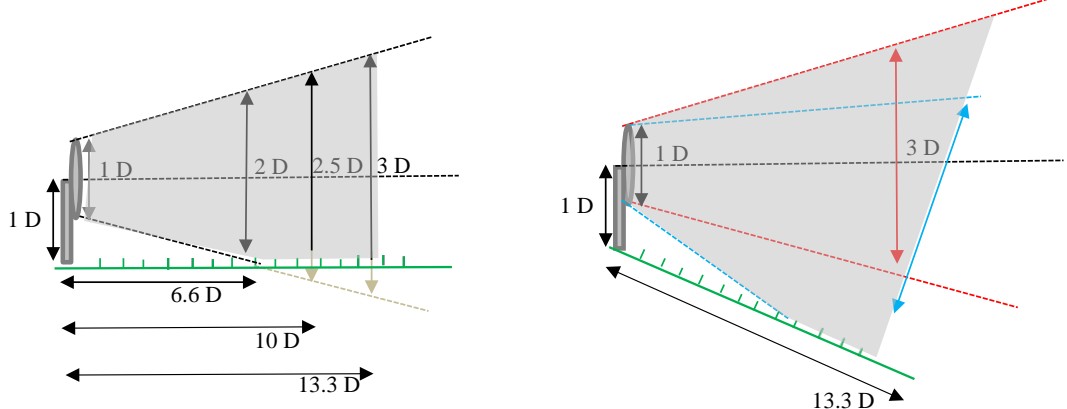

**Figure 2. Schematic of wind turbine wake expansion downwind and the scanning volume needed (grey shading) in near-neutral**
**conditions with the horizontal wake centreline shown by the black dashed line. Left: In flat terrain, for $k$ = 0.075 the wake can be expected to impact the ground at 6-7 $D$ downwind. Right: In complex terrain e.g. for a turbine placed at a ridge the wake can be terrain following (indicated in blue) or lofted (indicated in red) hence the scanning volume required in order to detect and characterize wakes is much larger.**





Wakes are advected by the ambient flow and thus, in addition to expanding, they are also subject to both horizontal and vertical meander (Larsen et al., 2008). Although many simple models assume the wake to be axisymmetric around the wind turbine hub-height (Jensen, 1983), both shear and veer in the inflow causes wakes to become highly asymmetric (Bodini et al., 2017;Barthelmie et al., 2016b) and this effect is amplified in stable conditions when the boundary-layer is shallower at least in flat terrain (Abkar
and Porte-Agel, 2016). Large eddy simulation results indicate that shape of the wake and the maximum velocity deficit moving downstream are also impacted by the Coriolis force (Abkar and Porte-Agel, 2016). While the majority of both analytical and numerical models broadly capture wake features such as wake velocity deficit (especially for single wakes) (Ainslie, 1988), it has proved difficult to improve predictive models that quantify wake details in large offshore wind farms (Barthelmie et al., 2004) or the net effect on the downstream atmosphere from large onshore arrays (Pryor et al., 2018) due both to wake complexity and a
shortage of available wake measurements for model evaluation.

Complex terrain typically has higher $z_0$ and thus higher ambient turbulence intensity which may lead to faster wake recovery, particularly if enhanced by unstable conditions (Han et al., 2018). The wake shape is modified by the movement of the wake downhill with the wake centre height following the slope but the location and shape of the wake is also modified by the turbulence intensity (Politis et al., 2012). The wake from the wind turbine will also interact with and be modified by the hill wake. It is likely
that the impact of the topography on the wake at the study site considered herein (Perdigão) is even more profound and complex than these previous studies due to the multi-scale topographic, thermal and canopy influences on flow and wake behaviour.

### 1.4    Remote sensing of flow and wakes

Remote sensing approaches, particularly application of lidars, are increasingly being leveraged to provide flow characterization for the wind energy industry (Risan et al., 2018;Mikkelsen et al., 2008;Berg et al., 2015) and can be used to characterize the 3-
dimensional wake volume as it evolves downwind from the wind turbine as well as providing concurrent freestream wind speeds from upstream measurements (Barthelmie et al., 2018;Barthelmie et al., 2014). Doppler lidar deployed for wake characterization can either be installed on the wind turbine nacelle, where they have been shown to be effective for characterizing individual wakes from 2-6 $D$ (Aitken and Lundquist, 2014), or on the ground relatively close to the wind turbine (Doubrawa et al., 2016) or at a distance scanning towards the wake(s) (Torres Garcia et al., 2017;Barthelmie et al., 2014). It is frequently difficult to get permission
to install a Doppler lidar on the nacelle, and there is often a desire to sample wakes from multiple wind turbines simultaneously, thus most field campaigns for quantification of wind turbine wake characteristics involve Doppler lidar placed on the ground as here  (Barthelmie et al., 2014;El-Asha et al., 2017;Iungo et al., 2013;Clifton et al., 2018). Most frequently used scanning patterns comprise; one or more arc scans (in each arc scan the scan elevation angle is held constant while the azimuth angle is varied) which can be used to derive pseudo Plan Position Indicator (PPI) information, and/or Range Height Indicator (RHI) scans (varying
elevation, fixed azimuth angle) and/or Vertical Azimuth Display (VAD) (high elevation angle, 360° scan at fixed azimuth angles). Determining the ideal location for scanning Doppler lidar and designing the scanning geometry to optimally sample the wind turbine wake(s) relies on detailed knowledge of the wind climate including prevailing wind direction (recall lowest uncertainty in retrieved radial velocities is achieved when the line-of-sight is aligned with the wind direction, and error increase with increasing angular offset (Wang et al., 2016)) and is also constrained by practical considerations such as the availability of a secure and
reliable electricity supply, security, data and personnel access and so on.

Subjectively identifying the presence of wind turbine wakes using data acquired from scanning Doppler lidar deployed in flat terrain is relatively straightforward, but using scanning Doppler lidar to objectively detect and track the wake centreline and quantify wake metrics is more challenging. Most research focussed on lidar-based wake characterization has employed case studies either because the field campaign is relatively short providing only a few cases (Banta et al., 2015) or because it is difficult to



automate the process of identifying and quantifying wake deficits, shapes, meander and so forth (Bodini et al., 2017). Adding to the challenges of multiple wake characterization, the behaviour of wakes in complex terrain is more difficult to both measure and model and important supporting information such as atmospheric stability and turbulence intensity that have direct and measurable effects on changes in power production due to wakes (Barthelmie and Jensen, 2010), are frequently only available from a limited

number of instruments deployed on meteorological masts (Machefaux et al., 2016).

## 2    The Perdigão field experiment

The current research reports measurements from the largest field experiment in boundary-layer meteorology undertaken to date at the complex terrain site Perdigão in central Portugal (Figure 1). It is part of a series of experiments undertaken within the New European Wind Atlas project designed to improve characterization of wind resources and operating conditions for the wind energy

industry using high-resolution, high-fidelity observations of boundary-layer flow from both traditional meteorological platforms and remote-sensing instruments such as ground-based Doppler lidars (Mann et al., 2017).

The Perdigão study area is dominated by two parallel ridges oriented northwest to southeast, separated by approximately 1.6 km (peak to peak), and with lateral extent of > 4 km (Figure 1). The ridges extend 300 m above the local terrain and about 170 m above the central valley. A single 2 MW Enercon E-82 wind turbine (hub-height ($WTHH$) of 78 m and a rotor diameter ($D$) of 82

m) is deployed on the southwest ridge (Figure 1). At this location the ridge has an approximate height ($H$) ~ 300 m and $L_h$=800 m. Slopes from the central valley towards the wind turbine have an average value of 16%, but are steep near the valley base (~30%), gentler in the middle portion and then increase to 40% and > 60% in the last few tens of meters at the ridge top (Figure 1). The land cover/canopy is also heterogenous with areas of grass/low vegetation in the valley transitioning to coniferous trees on the lower slopes and then mainly eucalyptus trees or bare areas/low shrubs close to and along the ridge. For the terrain specifications

of Perdigão and flow from the southwest (i.e. inflow for wind turbine and thus wakes that potentially will enter the Galion lidar scanned volume), the inner and middle layer heights of $l$~50 m (equation 1) and $h_m$~ 284 m (equation 6) for near-neutral stability (Table 1). This implies that at least part of the wind turbine wake will lie within the inner-layer and will be strongly impacted by the surface characteristics.

**Table 1. Inner-layer ($l$) and medium-layer heights ($h_m$) for southwesterly flow to the wind turbine at Perdigão (see Equations 1-7).**

|  | Near-neutral | Stable | Unstable | Near-neutral | Stable | Unstable | Near-neutral | Stable | Unstable |
|---|---|---|---|---|---|---|---|---|---|
| $z/L$ ($z$=78 m) |  | 0.16 | -0.16 |  | 0.08 | -0.08 |  | 0.16 | -0.16 |
| $z_0$ (m) | 0.3 | 0.3 | 0.3 | 0.3 | 0.3 | 0.3 | 0.1 | 0.1 | 0.1 |
| $L_h$ (m) | 800 | 800 | 800 | 800 | 800 | 800 | 800 | 800 | 800 |
| $l$ (m) | 50 | 46 | 64 | 50 | 50 | 63 | 42 | 37 | 52 |
| $h_m$ (m) | 284 | 261 | 262 | 284 | 227 | 240 | 266 | 220 | 232 |

The main Perdigão field experiment ran from mid-January to the beginning of July 2017. During the experiment sonic anemometers and other micrometeorological instruments were deployed at heights of 10 to 100 m on 50 fixed meteorological masts. In addition to the meteorological masts, multiple Doppler lidars and sodars were operated during the entirety of the main experiment and many other instruments were deployed during the Intensive Operating Period (IOP) from May 15-June 30 2017 (Fernando et al., 2018).

Meteorological conditions close to the wind turbine are summarized herein using 18 Hz data collected using a Gill WindMaster Pro sonic anemometer deployed at z = 78 m a.g.l. on Tower 20 (a 100 m meteorological mast) on the southwest ridge displaced approximately 180 m southeast of the wind turbine (see further details of the flow variability across the site given in (Letson et al., 2018) and (Fernando et al., 2018)). In the analysis of wake centreline behaviour under different prevailing atmospheric conditions



atmospheric stability close to the wind turbine is represented by the Monin-Obukhov length $L$ (equation 4) determined from the 78 m sonic anemometer measurements in each ten minute period and allocated to a stability class based on $z/L$ where $z/L<-0.08$ is unstable, $-0.08<z/L<0.08$ is near-neutral, $z/L>0.08$ is stable (Barthelmie, 1998). The inflow turbulence intensity ($TI$) in each 10-minute period is computed from the same sonic anemometer measurements as:

$$TI = \frac{\sigma_U}{U} \tag{12}$$

Wake detection for the entire period of operation of the scanning pulsed Doppler lidar operated by Cornell University (mid-January and the end of June 2017) are presented herein. However, data from Tower 20 are only available from March, thus the characterization of wake centre position as a function of prevailing meteorology can only be considered for March-June, inclusive. The Galion 4000 lidar has a wavelength = 1.56 μm, a pulse length of 30 m and a range of up to 4 km (Wang et al., 2015). The instrument was operated from 980 m northeast of the wind turbine at a location in the central valley (Figure 1). Pre-deployment

planning focussed on development of an optimal scanning geometry for the scanning Doppler lidar sufficient to acquire a data set to rigorously evaluate an objective processing methodology and to provide quantitative metrics of the location and characteristics of wind turbine wakes in complex terrain. The scan configuration described below is thus designed to permit continuous autonomous operation in the long-term, and balance having sufficiently high-density scans to permit identification of the wake (in both the time and space domains) while not defining too small an overall arc span that would preclude collection of a meaningful

number of cases. This measurement strategy was informed by the wind climatology for the site, and results from the test experiment Perdigão 2015 (Vasiljević et al., 2017) that indicated the streamline deformation at and downwind of the ridge is highly variable and associated with a wide range of wake behaviour including lofting and descending and terrain following. In the following Methods section, the creation of the scanning geometry and development of the automated processing algorithm are described.

## 3    Methodology for automated wake detection in complex terrain

### 3.1    Defining lidar scan geometry

For a scanning Doppler lidar with a relatively slow-moving head (such as the Galion used here) design of a scan geometry that integrates a combination of arc, RHI and VAD scans, requires consideration of a range of temporal and spatial factors including:

- To capture the wind turbine wake with a sufficient spatial detail and number of repetitions to derive statistically robust information requires assumptions regarding the prevailing wind direction in order to limit the arc span (i.e. range of arc
azimuth angles).
- Selecting low elevation angles for arc scans allows the lower portion of the wake to be observed as the wake expands and then potentially impacts the ground and also minimizes errors in transforming radial velocity into cartesian wind speed co-ordinates. For a Doppler lidar placed close to the wind turbine, selecting low elevation angles will not allow the wake top to be measured while choosing elevation angles that are higher will allow the beams to measure across the near-wake
but then will scan beyond the wake top after short distances. Assuming the Doppler lidar is placed at the turbine base for the case in Figure 2a, a scanning angle of 17.6° is required to scan to the top of the wake.at 6.6 $D$ and, assuming a standard expansion rate this would cover the top of the wake to 10 $D$ distance. For a Doppler lidar placed at a distance scanning towards the wind turbine, low elevation angles and a clear line of sight are needed. For example, after 1 km an elevation angle of 2° will already have attained a height of 35 m and an angle of 4° would have a height of 70 m. Hence to capture
the whole wake area in a vertical slice, careful planning is needed for proper selection of the elevation angles.





- Using a wide arc span can reduce uncertainties in flow characterization from Doppler lidar (Wang et al., 2016) and may capture a wider range of possible wake tracks, but increases the scan time thus decreases the possible temporal resolution.

- Reducing the number of range gates limits the maximum horizontal range, but decreases the dwell time for each individual azimuth and elevation angle affording the opportunity for decreased disjunct sampling duration (i.e. decreases the time between a specific volume being re-sampled)

- Selecting sufficient elevation angles is necessary to provide sufficient detail of the wake at various heights/distances (Doubrawa et al., 2017) but increasing the number of elevation angles increases the disjunct time interval.

- RHI scans can provide vertical slices through the wake as it moves downstream but if incorrectly aligned will produce 'empty' scans i.e. scans of the wind field without wakes unless wake tracking is employed (Wildmann et al., 2018).

- Vertical Azimuth Display (VAD) scans are useful for determining wind direction as well as providing a consistent time series at a range of heights, albeit for one location, but again inclusion of VAD scans can compromise inclusion of additional arc scans and/or lead to longer disjunct time increments.

Additional challenges for experiments in complex terrain include:

- If there is a need to retrieve wind speed components from the Doppler radial velocities generated from a single lidar it is necessary to assume homogeneity within the scanned volume (Wang et al., 2016), this criteria is unlikely to be realized in complex terrain (Pauscher et al., 2016).

- It is common practise to define wind turbine wake characteristics relative to a single freestream inflow profile and/or undisturbed downstream profile(s). While previous studies have invoked the concept that a wind turbine wake is a feature embedded in a logarithmic wind profile and have used that to define wind turbine wake characteristics (Aitken et al., 2014), vertical profiles in complex terrain with heterogeneous land use/land cover differ markedly from this assumption. However, deriving representative freestream wind speed profile(s) is difficult in complex terrain and it will also be more difficult to measure one or more upstream wind speed profiles, depending on the terrain and scanning Doppler lidar location. The horizontal and vertical complexity of the flow may mean that there is no one wind speed/turbulence intensity profile that is representative of inflow conditions across the rotor plane during a 10-minute period and this will result in increased uncertainty in the derived wake characteristics.

- Wakes in complex terrain exhibit a more diverse range of behaviours than in flat terrain (as illustrated in Figure 2a) because they are embedded in flow where there may be very strong shear and/or veer, and the streamlines may be closely terrain following or may exhibit vortex structures and zones of attachment/detachment. The scan geometry must be designed in a manner that can capture wakes that exhibit negative and positive vertical displacement from the original axis of generation. Accordingly, as indicated in Figure 2b, the scanned volume will need to be larger than in flat terrain. Further, it is likely that the turbulence intensity will be relatively high and wakes will not be preserved over long distances as they are offshore (Barthelmie et al., 2013).

These points serve to illustrate the issues in finding a scanning geometry with both sufficient spatial and temporal resolution. For Perdigão, the number of scan types employed for use in the scanning Doppler lidar is limited to optimize wake capture, ensure the disjunct sampling did not extend beyond 10-minutes (i.e. that the full scan is completed in 10-minutes) and reduce the complexity of the data processing. A schematic of the final version of the scan geometry designed for the Perdigão campaign is shown in Figure 1. It comprises a series of 10 fixed elevation arc scans with telescoping (i.e. variable intervals) in both the vertical and azimuth close to the direct alignment with the wind turbine. Elevation angles are; 9, 10.5, 12, 13.5, 15, 16.5, 18, 19.5, 21 and 23°. The arc angle in the azimuth varies with the elevation angle but for 15° comprises 23 beams from 199-259° with resolution between 1.5 and 6°. At the end of each scan 3-4 VAD scans (Azimuth angel: 0:30:360, Elevation angle: 56°) are included to allow an initial



estimation of direction and wind speed at a height equal to the wind turbine to be determined and provide a vertical profile of wind speeds within the valley for use in other research. Minor adjustments were made to the scan pattern in February 2017 to improve the efficiency of the scan including reducing the number of range gates to 41 making the maximum distance scanned 1230 m. In May 2017 the lowest elevation angle of 7° was removed because it was frequently blocked by vegetation and additional RHI scans

directly towards the wind turbine at ~ 227° (Az: 226.9 and 227.4°) were added for comparison with other lidar data sets from the IOP (Barthelmie et al., 2018;Wildmann et al., 2018). These modifications were kept to a minimum to ensure continuity in the subsequent data analysis.

### 3.2    Quantifying freestream conditions

A description of the freestream (i.e. inflow) conditions is critical to subsequent wake metric determination and also subject to high

uncertainty, particularly in complex terrain. For a single wake in flat terrain or offshore, the freestream wind speed can be assumed to be a single location wind speed immediately upstream of the wind turbine at the turbine hub-height or a simplified wind speed profile (Barthelmie et al., 2010). However, as rotor planes increase in size or as flow veers and shears in complex terrain, it becomes increasingly difficult to accurately describe the inflow profile. Although there is a meteorological tower (Tower 20) located 180 m along the ridge from the wind turbine there are issues with using this as the freestream due to the acceleration and turning of the

flow at the ridge and the spatial variability of the inflow wind field (Menke et al., 2018a). Further, the intention of this research is to develop a process in which a single scanning pulsed Doppler lidar could be deployed without additional instrumentation and operated autonomously to detect and characterize wind turbine wakes. Thus, an additional consideration for the scan geometry definition and data processing is to include a sufficient number of range gates to allow characterization of the freestream winds that impinge upon the rotor plane.

In the analyses presented herein the process for identifying potential wind turbine wakes cases that may have been sampled by the Galion lidar is multi-step. First, the approximate wind speed and direction at a height above the Galion lidar equal to the $WTHH \pm$ 30 m are estimated from the VAD scans as the maximum negative mean value of all valid radial velocities (where a signal to noise threshold (SNR) of 1.015 is applied) within each 10-minute period. If this analysis indicated a wind direction of 210 or 240°, the arc scan radial velocities (SNR > 1.015) at a range distance of the distance to the $WTHH$ +40 m and for scans at an elevation angle

of 12-17° are used to refine the estimated inflow wind direction and wind speed. A comparison of the wind speed and direction as derived from sonic anemometer data collected at 78 m on Tower 20 and the radial wind speed and direction calculated from arc scans for a height equal to the $WTHH\pm40$ m is shown in Figure 3. Although there is a consistent relationship between the independent measurements of wind speed and direction close to the wind turbine from the sonic anemometer and the lidar with, for example, a linear fit of wind speeds yielding an intercept of < 0.5 ms$^{-1}$ and a slope of 0.94, there is also considerable scatter.

There is less good agreement for wind direction (note the discretization of wind directions from the lidar is a function of the scanned azimuth angles). While some of the scatter apparent in Figure 3 may reflect fundamental differences in the observations from lidars and sonic anemometers (Wang et al., 2015), some fraction likely derives from heterogeneity in flow conditions along the ridge (see for example Figure 4 in (Vasiljević et al., 2017)). For this reason, the inflow wind speed and direction derived from the Galion lidar are used only to set a flag that indicates a wind turbine wake is likely to be present within the volume scanned by

the scanning Doppler lidar (i.e. wind speeds are above the cut-in for the wind turbine and the wake is likely to be propagated into the sampled volume).





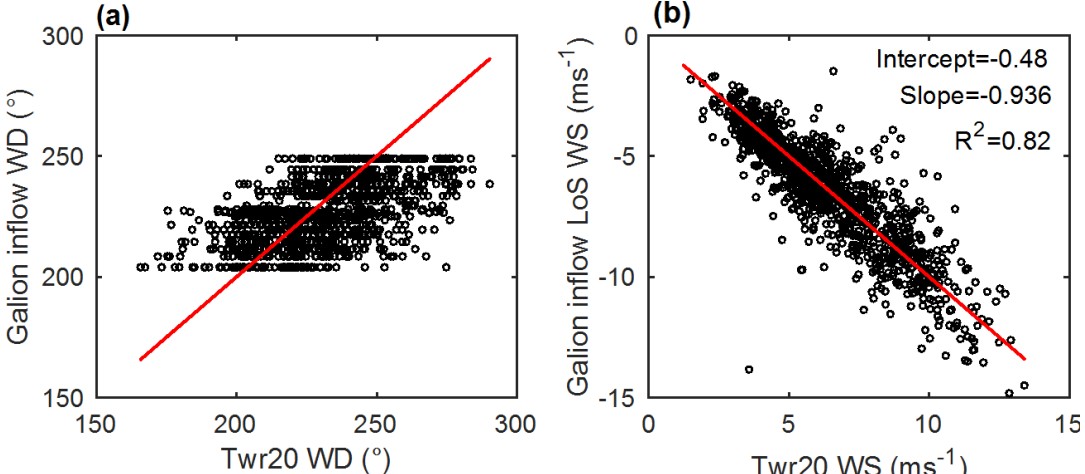

**Figure 3. Comparison of 10-minute (a) inflow wind direction and (b) inflow wind speed at 78 m height from scanning Doppler lidar measurements (labelled Galion inflow) and those from the sonic anemometer deployed at 78 m on Tower 20 (shown as Twr20). Note the wind speeds reported by the Galion are radial line of sight velocities and are negative for flow towards the wind turbine. 1:1 lines are shown in red while each dot represents a 10-minute period (n = 1123).**

### 3.3    Scan processing for identification of possible wake cases

As the volume of data sets being collected from remote sensing devices for wind turbine wake analyses increases there is a need to transition from manual analyses to case studies to automated procedures capable of generating statistically robust ensembles. The objective of the scan processing is thus to develop an algorithm that can detect the presence of wakes while rejecting non-wake cases and can derive quantitative wake metrics, including the focus here of centre location, with sufficient fidelity and detail that they can be used to describe the statistical properties of wind turbine wakes with increasing downwind distance from the wind turbine(s). In the following, radial velocities are given to avoid introducing errors/artefacts associated with the transformation into Cartesian coordinates. The objective is to create an algorithm that can detect and quantify wake features in long-term measurements, even in complex terrain where the minimum velocity is not always observed in the wake and the flow is affected by recirculation (Menke et al., 2018b).

As shown in Figure 4, an algorithm is developed and applied to data from Perdigão that starts with assessment of the status and fast processing of the VAD scans for an initial determination of the possibility of wakes. It would naturally be useful to incorporate SCADA data to ascertain the operational state of the wind turbine but these are not available. If the case passes this criterion an evaluation of the direction and radial wind speed from the arc scans made close to the wind turbine location to determine whether a wake is likely to be present and whether it is being advected in a direction sampled by the arc scan volumes (see an example of the arc scans in Figure 5). Then for each of the downstream distances considered (i.e. the vertical planes located at 2 $D$, 2.5 $D$, 3 $D$, 3.5 $D$, 4 $D$, 4.5 $D$) an assessment is made of whether there are sufficient retrieved radial wind speeds (i.e. measurements with a SNR > 1.015) to describe both the presence of a wake and the background flow field for each downstream distance and height.

Once each case has passed these filters, radial wind fields on these planes are used to derive anomaly fields (see section 3.4) from which the wake centre location is identified and other wake metrics are calculated for each downstream distance using both original data and cubic-spline interpolated fields. Hence this dataset and the wake detection is entirely self-contained. The performance of the algorithm in determining the wake centreline location is discussed in the Results section.



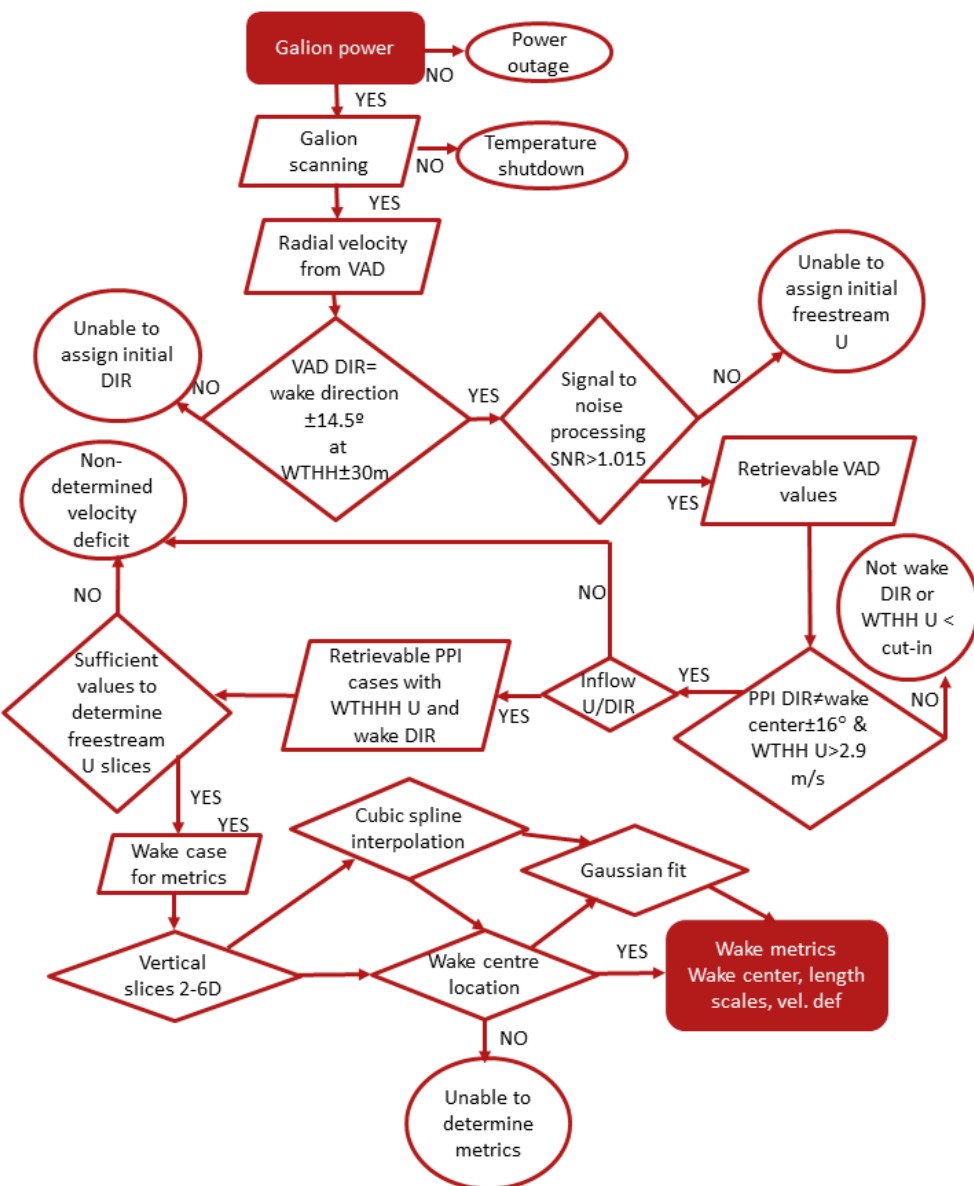

**Figure 4. Flow chart outlining the automated wake processing of the Doppler lidar data for wake detection and characterization.**

### 3.4     Wake characterization

One of the main issues in wakes research is in developing objective techniques to identify and quantify wake features (Aitken et

5  al., 2014). Most previous field research has been in relatively flat terrain (Smith et al., 2013) or offshore (Barthelmie et al., 2013)
and even in these environments it is very challenging to characterize wakes quantitatively (El-Asha et al., 2017). Understanding
the background flow field in which the wake(s) is(are) embedded is critical to defining spatial and temporal variability of wake





length scales, velocity and shape evolution with distance (Barthelmie et al., 2010;Doubrawa et al., 2016). In future work quantitative methods will be developed and applied to the full set of wake measurements from our complex terrain site in order to find robust yet sensitive methods that can be applied routinely to quantify wake characteristics. Herein, we focus on automated identification of the wake centre location and use of the automated detection algorithm to characterize variations in the wake

trajectories as a function of the prevailing meteorology.

As discussed above, a description of the undisturbed flow field is necessary for use in identifying the location of the wake centreline and defining wind turbine wake characteristics such as the normalized velocity deficit ($v_d(x,y,z)$):

$$v_{d(x,y,z)} = \frac{U_{0(x,y,z)} - U_{(x,y,z)}}{U_{0(x,y,z)}} \tag{13}$$

where $\Delta U$ is the velocity deficit and $U_0$ is the background flow field velocity. Defining both the $U_0$ and $U$ in the x, y and z directions is the primary challenge to wake detection and characterization in complex terrain. The approach adopted here is to define these

properties on planes at six fixed downstream distances from the wind turbine (i.e. y is set at 2 $D$, 2.5 $D$, 3 $D$, 3.5 $D$, 4 $D$, 4.5 $D$). At each of these downstream distances (X $D$ ± 20 m) the radial wind speeds at each x location (lateral displacement distance from a direct transect to the wind turbine) and z height (where z is defined from the elevation of the wind turbine hub-height) are retrieved for each 10-minute period. Then the plane of radial velocities is discretized into 20 m horizontal planes and a mean radial velocity is computed. Anomalies from that 'background' profile are then used in the wake centreline identification. The location

of the wake centre is determined using maximum velocity deficit anomaly for each height starting its search at the expected location (*WTHH*) and refining that location by checking it moving around the grid cells in each horizontal plane. The algorithm assumes the wake has moved further downstream than the immediate rotor plane after the double bell wake shape is expanded into a near Gaussian shape) (Barthelmie et al., 2003) and hence that there is a well-defined centre.

To aid in understanding the wake detection algorithm the arc scan-derived radial velocities at the various elevation angles along

with the VAD scan for an example 10-minute period are given in Figure 5. As shown, the wind turbine wake is clearly evident to and beyond distances of 4.5 $D$ from the turbine in the arc scans with elevation angles 12 to 21° as reduced radial wind speeds. Also shown are the radial wind speeds from the multiple VAD scans (last panel) that are used to provide an initial wind speed and direction for a height equal to approximately the *WTHH* that are used as an initial estimate of the likely presence of a wake in the scanned volume and thus to start the processing. The inflow conditions as reported in the upper right of Figure 5 indicate the initial

estimate of inflow wind speed upstream of the wind turbine as determined from the arc scans. Figures 5-7 also illustrate the need for normalization of the flow field on each plane to generate anomaly fields (where the background flow field is discretized in the vertical) for automated detection of the wake location. As shown in Figure 6, the variation of raw radial velocities across the x-z planes at the various downstream distances is dominated by variability induced by the topography. Specific to this example, there is clear evidence of weak upslope flow at low heights above the ground, overlain by flow towards the lidar aloft. Thus, although a

wind turbine wake is evident it is difficult to identify unambiguously because other local minima of equal (or greater) magnitude are present due to other mechanisms. The complexity of this flow, and the deviation from typical logarithmic profiles, is also evident from the mean background profiles shown in the middle panels. Once the anomaly fields are generated by subtracting the mean background line of sight flow profiles (LoS(z)) at each downstream distance (i.e. each plane), the wind turbine wake centre is considerably more apparent and can be correctly identified and tracked by the automated algorithm (cf. Figure 6 and 7).



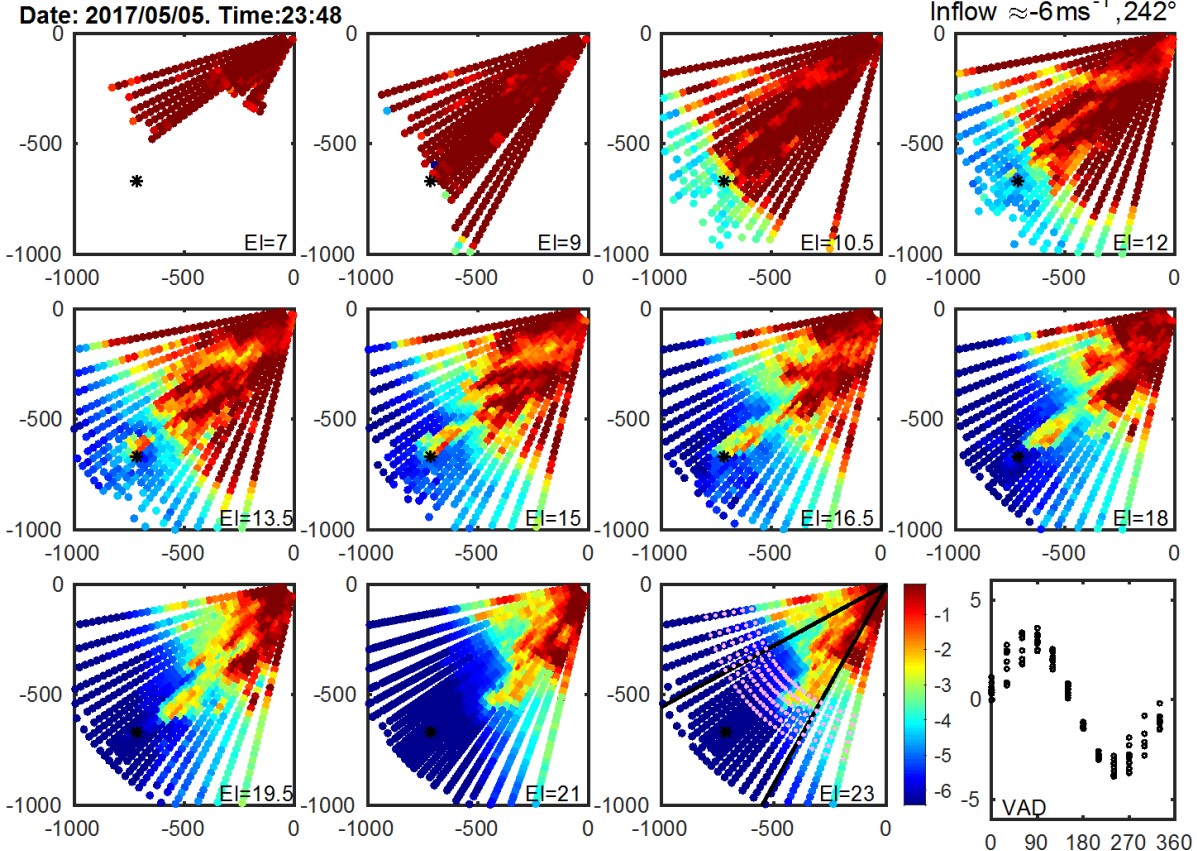

**Figure 5. Example of arc scans from May 5 2017 at 23:48 (UTC). During this 10-minute period the sonic anemometer at 78 m a.g.l. on Tower 20 indicated a wind speed of 5.9 ms⁻¹ and a wind direction of 252º, stable stratification with $z/L$ = 2.27 and $TI$ = 0.05. Data from the VAD scans (lower) indicated a wind direction of ~240 º. Conditions upstream of the wind turbine as characterized by the arc scans are 6.0 ms⁻¹ and 242º, respectively. Each panel is 1000 m by 1000 m with the Galion located at (0,0) and depicts line of sight (radial) wind speeds from each elevation angle; 7, 9, 10.5, 12, 13.5, 15, 16.5, 18, 19.5, 21 and 23°. The wind turbine location is shown by the black \*, and penultimate panel shows the location of the slices used in the wake characteristic quantification are shown as the magenta arcs at 2, 2.5, 3, 3.5, 4 and 4.5 D downstream of the wind turbine as well as the wake directions 209-241° by the thick black lines.**



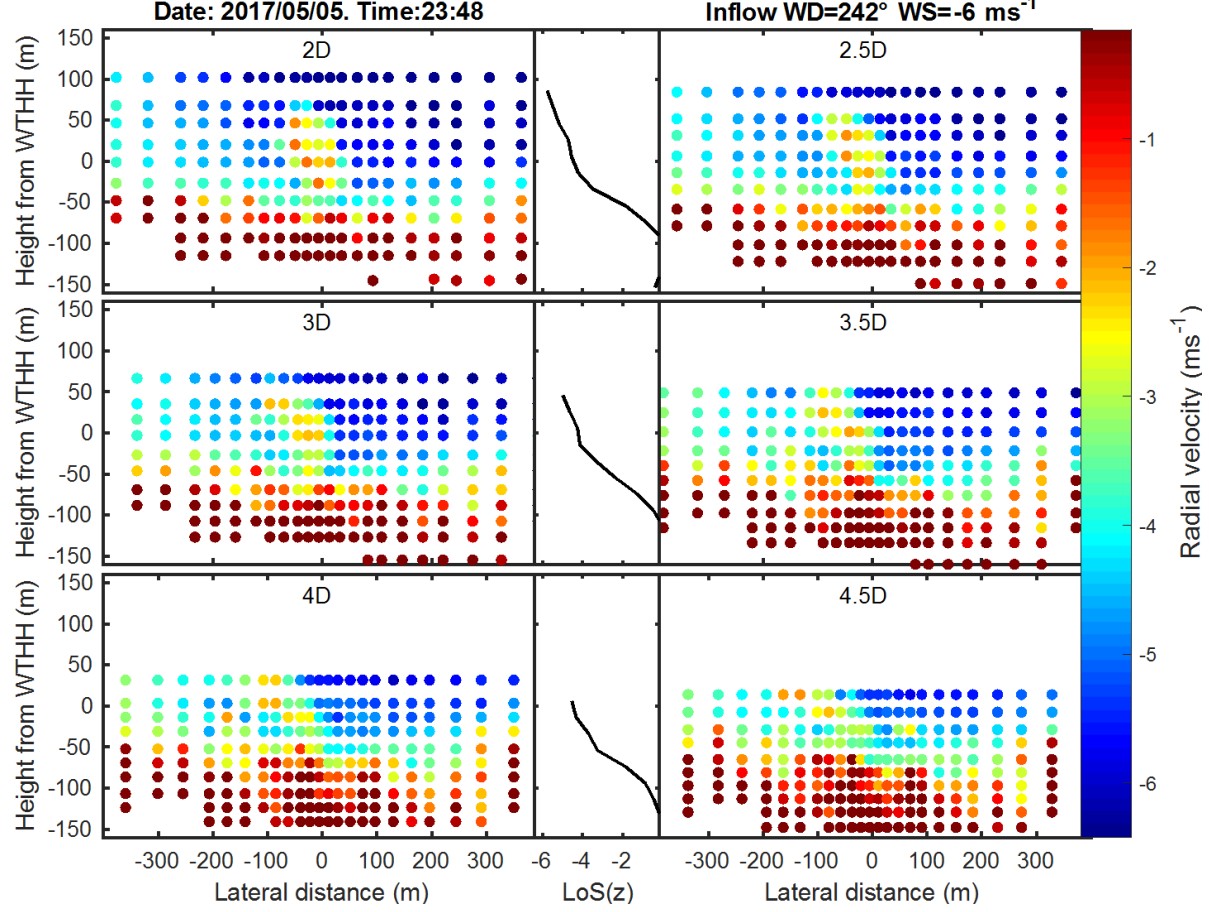

**Figure 6. Example of raw radial wind speeds on planes at the six downstream distances 2, 2.5, 3, 3.5, 4 and 4.5 *D* from the arc scans on May 5 2017 at 23:48 (UTC) (shown in Figure 5). The middle three panels show the mean background radial wind speed profile (LoS(z)) at 2.5, 3.5 and 4.5 *D*. A lateral distance of 0 (on the x-axis) indicates a direct line to the wind turbine (i.e. a direction of 227°). The heights on the vertical axis have been normalized to be 0 at the *WTHH*.**



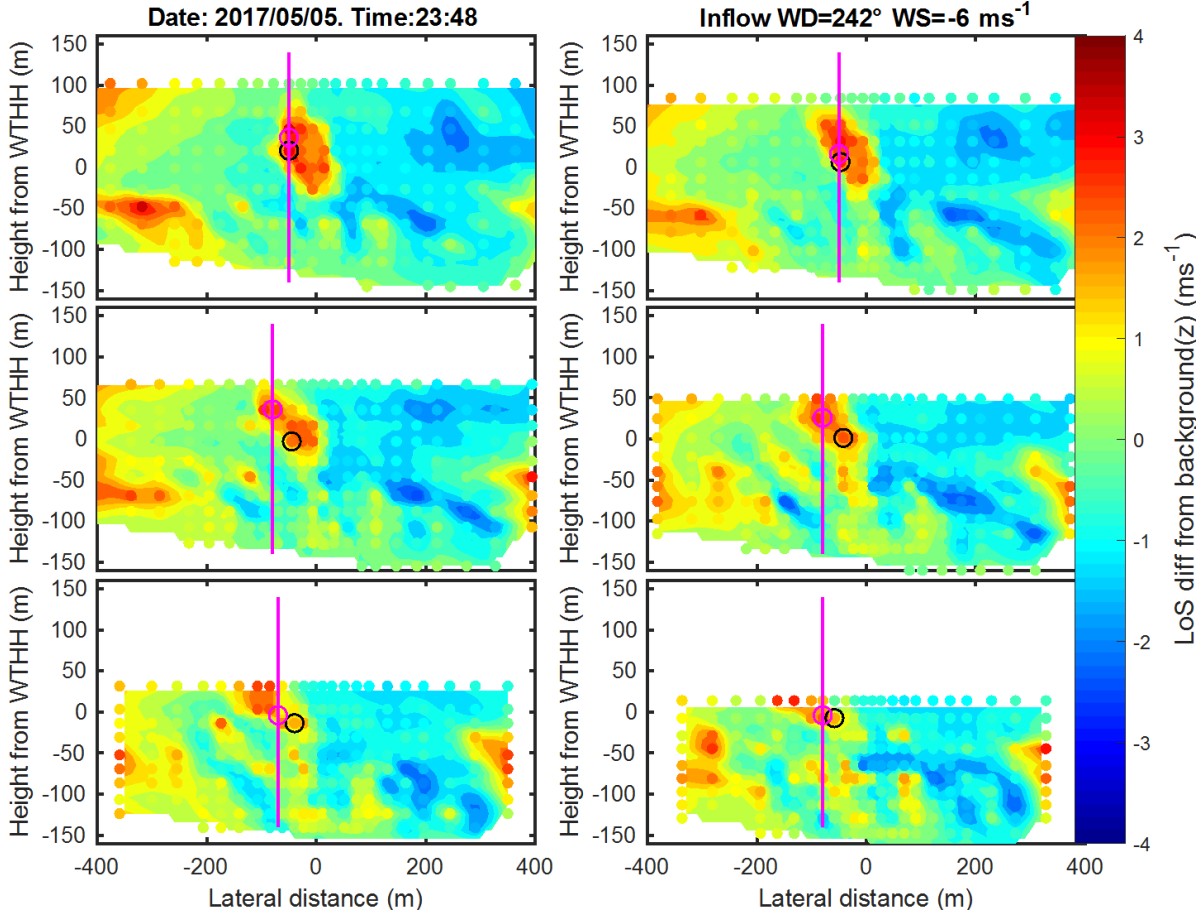

**Figure 7. Example of radial wind speed anomalies at the six downstream distances 2, 2.5, 3, 3.5, 4 and 4.5 *D* from the arc scans on May 5 2017 at 23:48 (UTC) (the data used to construct the anomalies are as shown in Figure 6). Points indicate the measurement locations while the background shading is a cubic spline interpolation of the anomalies (shown only to aid visual interpretation). The black open circle denotes the first guess location of the wake centreline from the previous 'slice' or in the first panel from the inflow conditions. The magenta circle shows the centre location as identified by the tracking algorithm. To aid in identifying the location of this circle a vertical line is show the bisects this location.**

## 4    Results

### 4.1    Meteorological conditions during the experiment

The Perdigão region was selected for the field experiment in part due to the presence of two approximately equal height ridges but also for the prevailing bimodal wind direction which means the flow is frequently oriented perpendicular to the ridges (northeasterly/southwesterly). This flow pattern is also observed during the measurement period (Figure 8). Higher wind speeds are observed during March and April but the high frequency of northeasterly winds during April meant relatively few wakes could be observed, despite the prevalence of wind turbine operating wind speeds (cut-in wind speed ~ 3 ms$^{-1}$ at *WTHH*). During May and June, wind speeds are consistently lower but still mainly above cut-in (Figure 8) and the frequency of winds from the southwest increased providing a higher number of wake situations that are detected by the Galion lidar.

In the absence of other alternatives, the stability classification and turbulence intensity from Tower 20 are used as an indicator of inflow conditions to the wind turbine. *TI* at approximately *WTHH* is most commonly between 0.1 and 0.2 for all wind directions.



The overall median is 0.13 ($25^{th}$ percentile = 0.09, $75^{th}$ percentile = 0.21, mean=0.17). The exception is northeasterly directions (from 50-70°N) when flow comes across the valley to the sensor, *TI* is generally lower with no observations above 0.3 and a large fraction below 0.1 (Figure 9). For the wake directions used here (209-241°), the high level of *TI* implies that wakes will generally recover rapidly (at short downwind distances) and may be strongly temporally varying. The stability classification identifies very

5 few observations (<5%) as being close to near-neutral (-0.08<$z/L$<0.08) with the majority of observations split between the stable (48%) and unstable classes (47%) with little directional dependence in the frequency of these classes (Figure 9). Instead, atmospheric conditions show a strong and expected diurnal cycle for a land site (Stull, 1988) with both the number of unstable conditions and the turbulence intensity increasing during the daylight hours (Figure 10). There is also clear association of much lower turbulence intensity with stable conditions overnight. The diurnal pattern of mean wind speeds is more complex with a

10 maximum in the early morning around 06:00 UTC (before sunrise at 07:52 in January to 06:02 in June) which might indicate the presence of a nocturnal jet and minimum values in the mid-morning around 10:00. This implies that the flow is impacted by both the valley and larger-scale synoptic systems and that complex wake behaviour is to be anticipated.





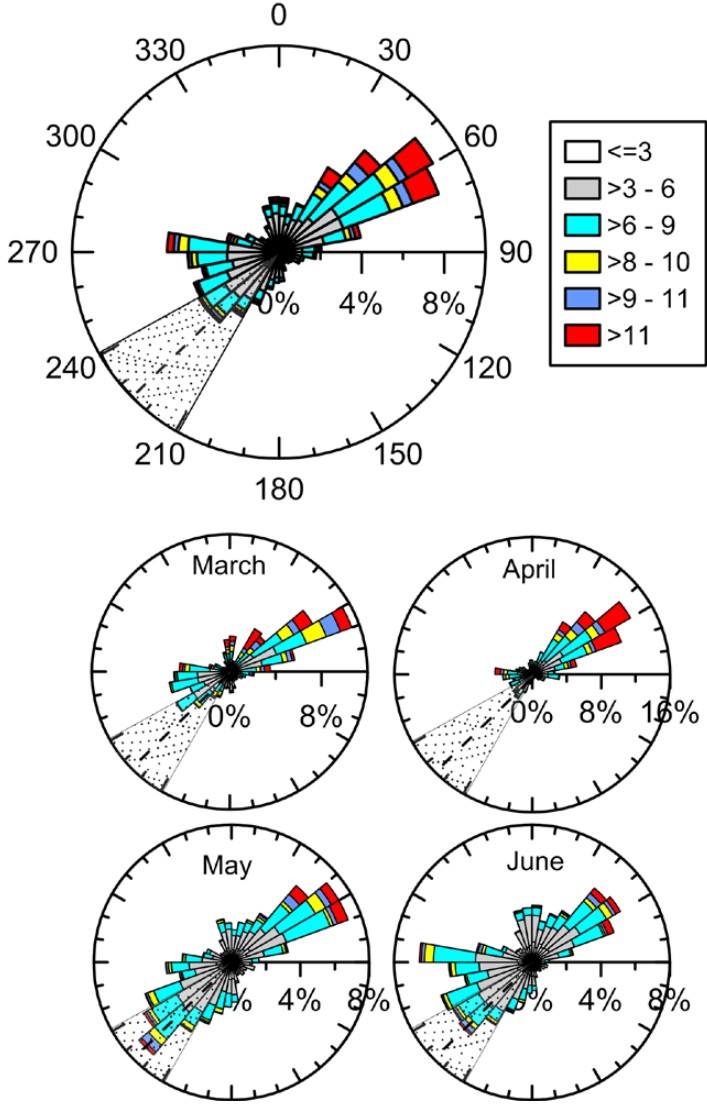

**Figure 8.** Top: Wind rose from 78 m height at Tower 20 (location shown Figure 1) for all 10-minute periods during March-June 2017. Coloured bars show the wind speed in ms⁻¹. Below: Wind rose for each month. The shaded grey area shows the primary wake directions (209-241°) with the wake centreline direction of 227° marked with the black dashed line.





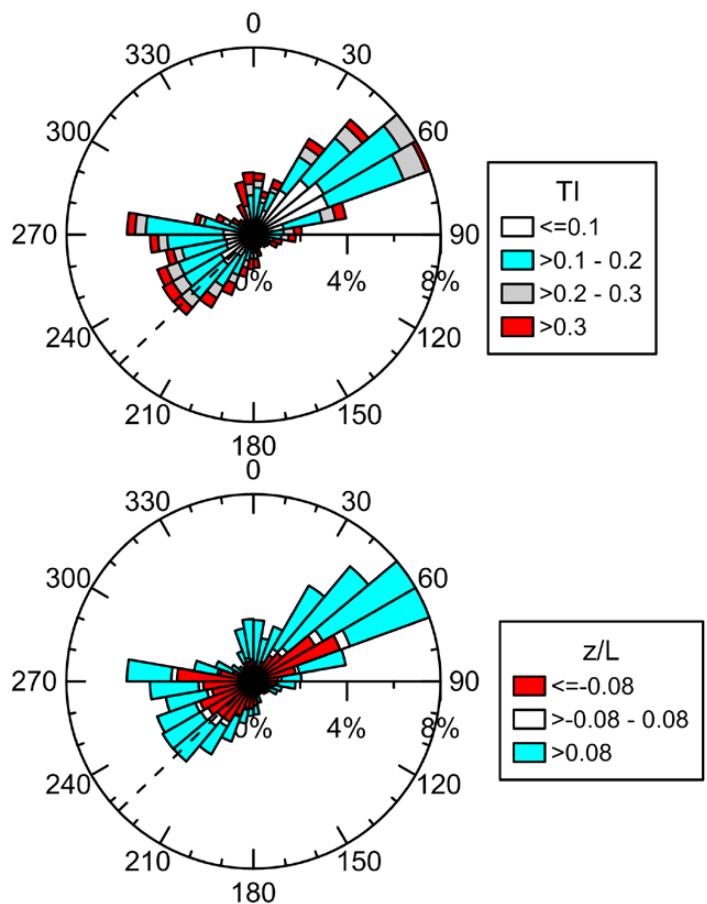

**Figure 9. Top:** Turbulence intensity rose from 78 m height at Tower 20 (location shown Figure 1) for March-June 2017. Coloured bars show the turbulence intensity defined from equation 12. **Below:** Atmospheric stability rose from 78 m height at Tower 20 (location shown Figure 1) for March-June 2018. Coloured bars show the value of *z/L* (*L* is defined from equation 4) where white is near-neutral, red is unstable and cyan is stable. The wake centreline direction of 227° is marked with the black dashed line.





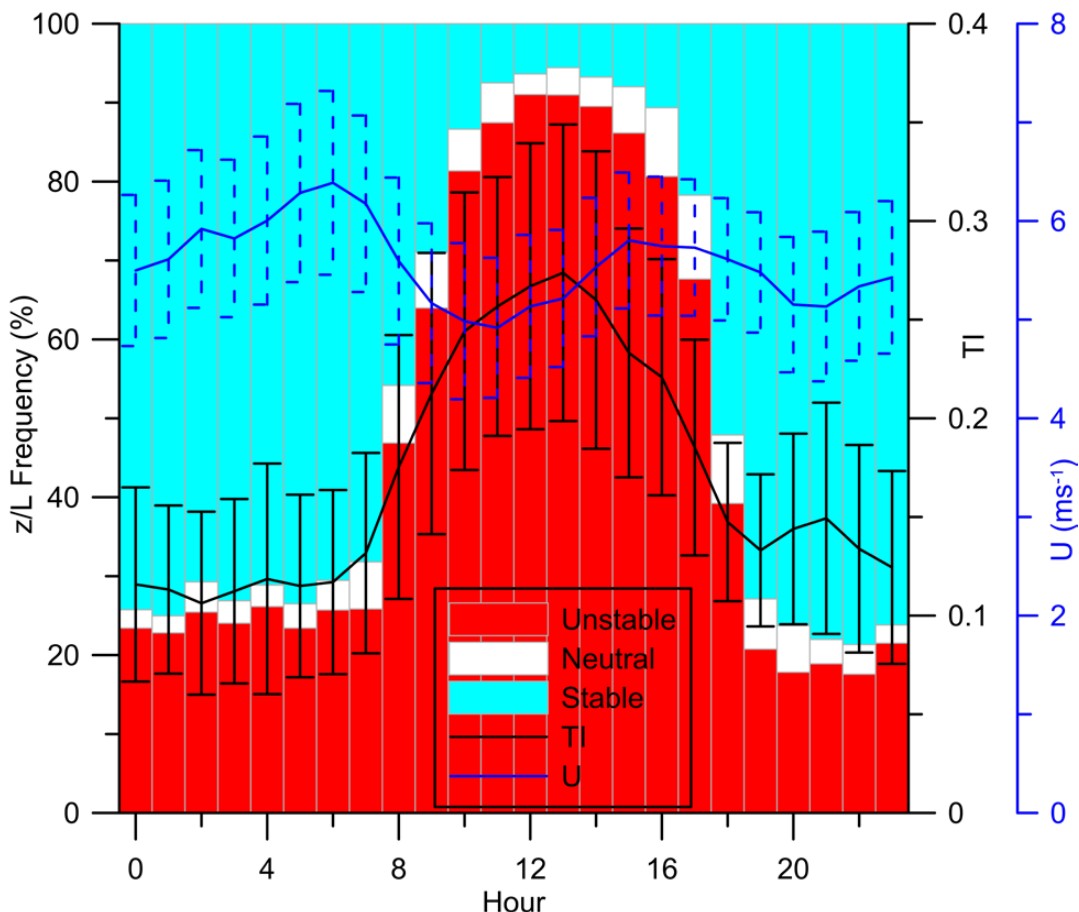

**Figure 10. Diurnal variability of meteorological variables from measurements at 78 m at Tower 20 for the whole data set (March to June 2017). The frequency of atmospheric stability classes ($z/L$<-0.08 unstable, $z/L$<|0.08| neutral, $z/L$>0.08 stable) (bars), wind speed ($U$, blue line with dashed error bars showing 0.25*standard deviation) and turbulence intensity ($TI$, black line with error bars showing 0.25*standard deviation).**

## 4.2    Data processing methodology

There were very few power or instrument issues during the measurement period of mid-January to end of June, leading to 92% availability of the scanning Doppler lidar instrument. Electrical power was supplied specifically for the experiment and most instruments including the Cornell University scanning Doppler lidar were placed on the temporary power grid and part of the availability reduction arose from a faulty fuse in an extension line. Before this was corrected, during March about 280 hours of potential scan time were lost due to power issues but only about 35 hours in April. As temperatures increased, the scanning Doppler lidar started to experience thermally-induced automatic instrument shutdown during the afternoons when ambient temperatures exceeded 30°C and wind speeds in the valley were very low. During May there were five individual shutdown events, accounting for 54 hours of missing scans in total. During June, afternoon shutdowns were routine occurring on more than half of days between the times of 10:00-20:00 (UTC) for an average of seven hours each. However, the instrument automatically restarted once its internal temperature dropped below 40°C and this provided for excellent data coverage and a database of 19,384 complete scans, each covering a 10-minute period.





An automated processing algorithm is necessary to remove subjectivity in wake characterization and hence aid reproducibility. The algorithm developed here is summarized in Figure 4 and the results of the pre-screening for potential wake cases are summarized in Figure 11. The requirement for a sufficient number of points in the VAD scan to calculate an initial wind direction (and wind speeds at *WTHH* sufficient for wake generation) removed very few cases (8) from further consideration. A large number

of 10-minute periods (15894) are excluded from consideration as potential wake cases because the wind direction as determined by the VAD scans falls outside the 209-241° direction range (labelled DIR). This is consistent with observations at Tower 20 that indicate a very high prevalence of north-easterly flow (Figure 8). A small number of possible cases (484) are identified as potential wake cases in data from the VAD scans but excluded from further consideration because the arc scans used to infer inflow wind speed and wind direction at the wind turbine either report too few observations at that range from the lidar with sufficient SNR to

merit inclusion in estimating the inflow wind speed (label INFLOW). A few further cases are excluded because the inflow wind speed derived from the arc scans are below wind turbine cut-in wind speed (U<3 ms$^{-1}$) (1027). These pre-screening selection criteria reduce the number of observations leaving 1971 cases (10.2% of the original scans) that are processed to derive quantitative estimates of the wake position (and characteristics). Naturally the criteria applied are highly selective but the purpose here is to identify cases from which quantifiable wake characteristics can be determined.

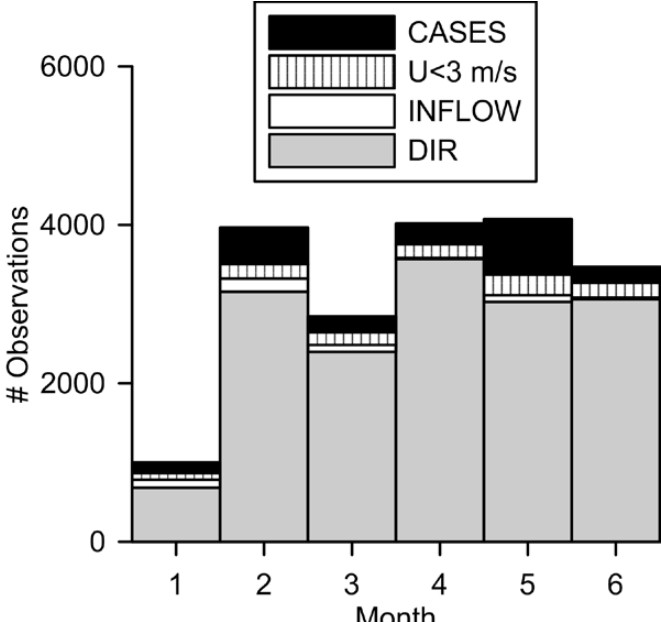

**Figure 11. Results of the wake case selection pre-processing (flow chart shown in Figure 4) by calendar month (1=January 2017). The total column heights indicate the number of 10-minute periods for which data are available in each month. DIR indicates the number of scans excluded from consideration because the VAD-derived wind direction was not 210 or 240°. INFLOW indicates the number of scans filtered out by having insufficient observations at or slightly beyond the wind turbine to determine the inflow. U<3 ms$^{-1}$ indicates**
**scans filtered by having wind speed at hub-height lower than the wind turbine cut-in wind speed at the turbine location. CASES denotes the 10-minute periods that are identified as potential wake cases that meet the criteria for quantitative processing.**

### 4.3    Evaluation of the wake centreline detection

Development of objective methods with which to evaluate the wake detection algorithm is challenging. To evaluate the performance of the automated detection algorithm an assumption is made that no wake cases are missed which is based mainly on

the practical consideration that wind speed and directions selected include all scans where the wind turbine could be operating and where the wake can reasonably be tracked for up to 4.5 *D* (see the last panel of Figure 7). Of the 1971 potential wake cases, only





one is rejected by the automated algorithm to locate a wake centre location at 2.5 $D$, surprisingly this number did not increase above three with increasing distance from the wind turbine. Subjective inspection of the wake cases indicated that they can be classified into four possible scenarios; A) straightforward wake cases with a clearly and correctly defined centre, B) cases where the wake centre is split but the wake centre of gravity is correctly identified by the algorithm,  C) cases where the algorithm is mis-

specifies the wake centre location and D) cases where the algorithm is unable to locate a wake but subjective inspection could not do so either (Table 2). Figure 12 shows illustrative examples of the four wake types on planes 2.5 $D$ from the wind turbine selected from a 24 hour period during 3-4 May 2017. These serve to illustrate why the objective process is challenging. For the A type cases the wake is shown as a central area where there is positive radial velocity with lower or negative velocities around it. Recall in all the cases presented, the wind direction is southwesterly and so the flow is towards the lidar (i.e. negative radial velocities)

and thus the wake will appear as a positive anomaly. These 519 events comprise 46% of all potential wake cases. Only a relatively small number of case (58) are subjectively identified as exhibiting a clear wake but that location is incorrect in the tracking algorithm as 2.5 $D$ (i.e. category C).  Not all wind turbine wakes are manifest as a single positive velocity anomaly. In these B cases, the wake area is present as multiple lobes and does not have an obvious wake centre. In 16% of the 1971 potential wake cases exhibit a split wake of this type (type B) but the algorithm correctly identifies part of the wake. In 33% of potential wake

cases no wake is subjectively observed although conditions would predict they should be present or it is not possible to distinguish wakes against the complexity of the background flow (Type D, Figure 12d). The example of a case D type shown in Figure 12d is very typical of the flow complexity with weak upslope/downslope flow to the right/left of the centreline to the wind turbine (shown as lateral distance =0). This flow pattern persisted for many consecutive time periods and thus appears to represent micro-scale topographic forcing of the flow (see slope variability in Figure 1). Naturally, not all case D wake types are reflective of flow

complexity, use of SCADA data might remove some of these cases as it is possible that the wind turbine was not operating during all 10-minute periods.

The diversity of possible outcomes under conditions that should be conducive to wake generation (Figure 4), also indicates how the process of developing a wake processing algorithm can be challenged by flow complexity. While data from both Tower 20 and the scanning Doppler lidar indicate flow from the southwest and a wake can clearly be identified for at least 4.5 $D$ downstream of

the wind turbine for the higher elevation scans, under the frequently occurring condition of recirculation flow in the valley (Menke et al., 2018b) (with radial flow away from the lidar while flow on the ridge is towards it), wake velocities blend with the local flow and cannot be distinguished, even if the wake is present. To assist understanding of the reasons why different wake forms (cases) wakes develop in different ways, the external conditions (stability, wind speed, wind direction and turbulence intensity) associated with each case are investigated (Table 2 and Figure 13). For Type A, the majority of cases occur in stable conditions (median $z/L$

=0.22) and with similar $TI$ (0.10) to the average (0.11), the mean wind direction is also very close to the direct centreline to the wind turbine, otherwise there is little to distinguish external conditions from the other cases in terms of the radial wind speed (-6.7 ms$^{-1}$). External conditions for type C are similar to type A but exhibit a greater frequency of highly stable conditions (median $z/L$=0.35) and a larger average directional offset from the direct azimuth angle to the wind turbine (4.1°). Split or multiple wake centre cases (type B) are associated with the highest radial wind speeds at the wind turbine (-7.5 ms$^{-1}$), the largest average

directional offset from 227° (7.8°), and are more likely to occur under unstable conditions (median $z/L$=-0.21) and high $TI$ (mean = 0.13). Type D cases are observed in atmospheric conditions that are similar to those that prevail during type B cases.





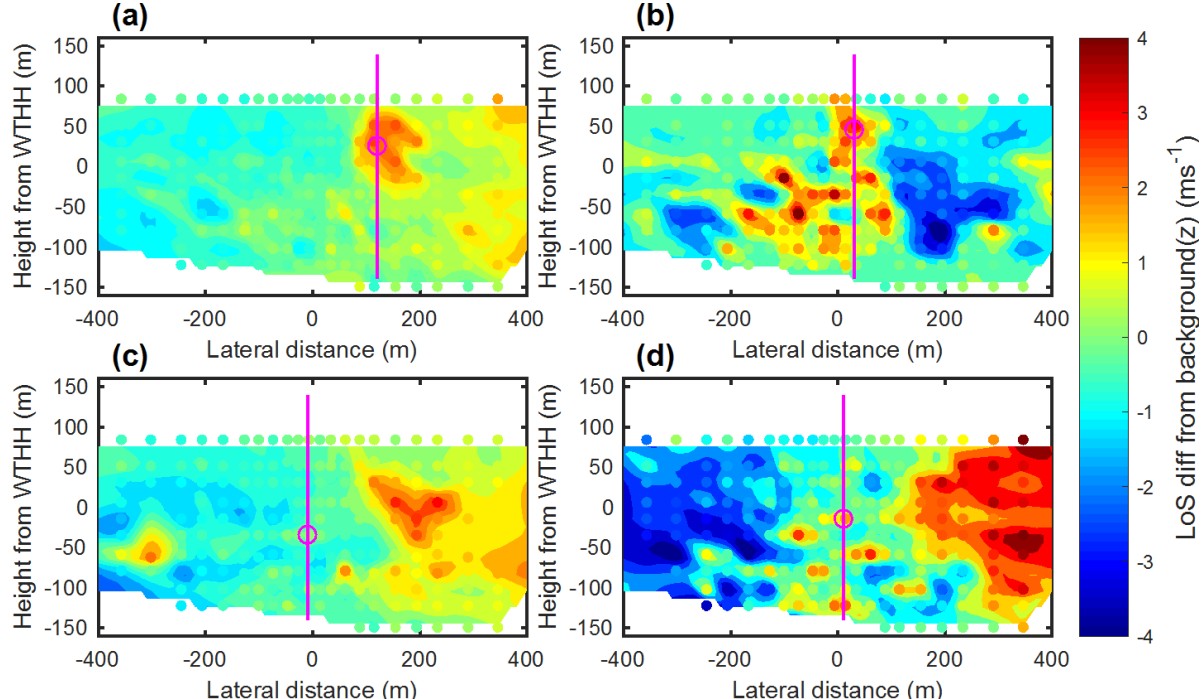

**Figure 12.** Representative examples of radial velocity anomalies at 2.5 *D* distance from the wind turbine for the four case types. The scanning Doppler lidar measurements are shown are individual points whereas the solid colours are cubic spline interpolated values. The results of the automated wake detection algorithm are shown as magenta circle for the wake centre. To aid visibility the location of the wake centre is extended vertically as a magenta line. The wake case types are;

**A. Clear wake centre, wake location correctly identified by algorithm**

**B. Split wake with no well-defined centre or multiple centres, location correctly identified by algorithm**

**C. Clear wake centre, location incorrectly identified by automated algorithm**

**D. Not possible to identify a wake centre subjectively or automated algorithm**

**Table 2.** External conditions from 78 m height at Tower 20 calculated for subjective classification of wake cases. There are 1971 wake cases in total, 1120 with meteorological data available.

| Wake category | # | Frequency (%) | Stability frequency (%) | | | Mean | Mean | Mean | Median |
| | | | Unstable ($z/L$<-0.08) | Neutral (\|$z/L$\|<0.08) | Stable ($z/L$>0.08) | Radial wind speed (ms$^{-1}$) | Direction offset from wake centreline (\|Dir-227\|) (°) | *TI* | $z/L$ |
|---|---|---|---|---|---|---|---|---|---|
| A. | 519 | 46 | 31 | 10 | 59 | -6.0 | 1.67 | 0.10 | 0.22 |
| B. | 179 | 16 | 56 | 18 | 26 | -7.6 | 7.76 | 0.13 | -0.21 |
| C. | 58 | 5 | 31 | 9 | 60 | -6.3 | 4.07 | 0.09 | 0.35 |
| D. | 364 | 33 | 55 | 8 | 37 | -6.5 | 5.08 | 0.13 | -0.21 |

The wake case types are;

**A. Clear wake centre, wake location correctly identified by algorithm**

**B. Split wake with no well-defined centre or multiple centres, location correctly identified by algorithm**

**C. Clear wake centre, location incorrectly identified by automated algorithm**

**D. Not possible to identify a wake centre subjectively or automated algorithm**





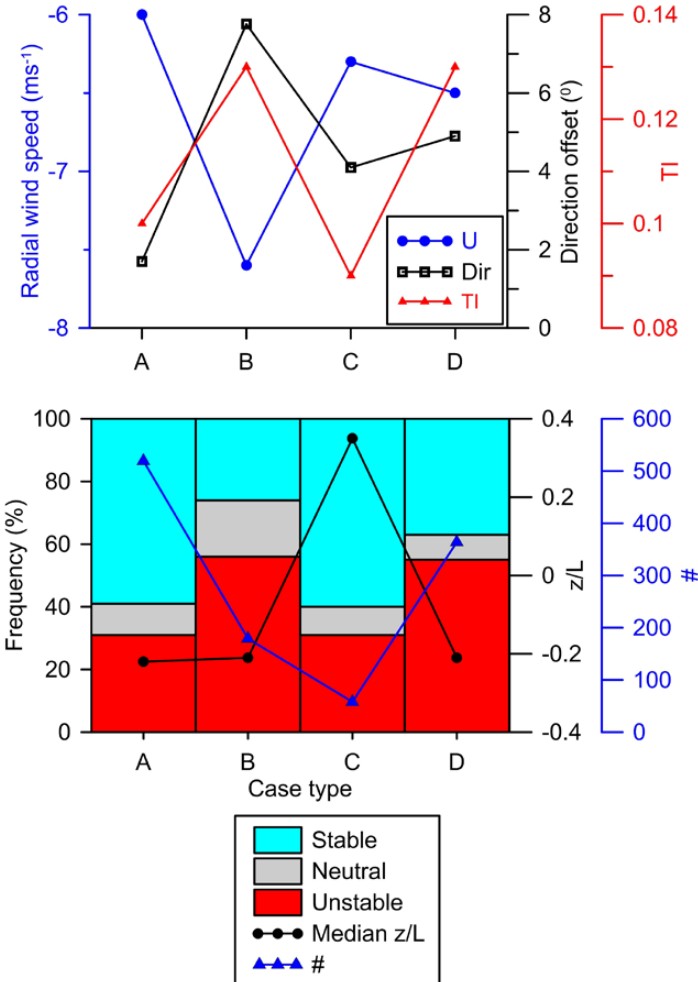

**Figure 13.** Mean inflow conditions and stability conditions conditionally sampled by wake class type where A=Clear wake centre, location identified, B=Split wake with no well-defined centre or multiple centres, location identified, C=Clear wake centre, location missed, D=Not possible to identify a wake centre. Stability is classified based on $z/L$ from 78 m height from Tower 20 where stable $z/L>0.08$, unstable $z/L<-0.08$ and neutral $z/L <|0.08|$. **Top:** Mean radial wind speed, turbulence intensity and direction from 227° by wake classes. **Bottom:** Frequency of stability conditions in each wake classes and the median value of $z/L$. Lines joining the classes are to aid clarity.

### 4.4    Dependence of wake centre location on prevailing external conditions

The 519 A type wake cases when sonic anemometer data are available from Tower 20 are conditionally sampled by wind speed and direction, stability (hour of the day) and turbulence intensity to examine the mean wake propagation characteristics and their dependence on the prevailing meteorology. At 2 $D$, the ensemble mean height of the wake centre is +11.1 m relative to the *WTHH* (i.e. as if the wake propagated on a horizontal line from the wind turbine centre), but subsequently it moves downslope, such that the average height at 4.5 $D$ downstream it is -36 m, or just under 0.5 $D$ below the *WTHH* (Figure 14). Although the standard deviation of the wake centre height at each downstream distance is large, the tendency of the wake centre to initially loft and then move down the slope is clear.

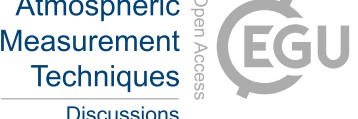



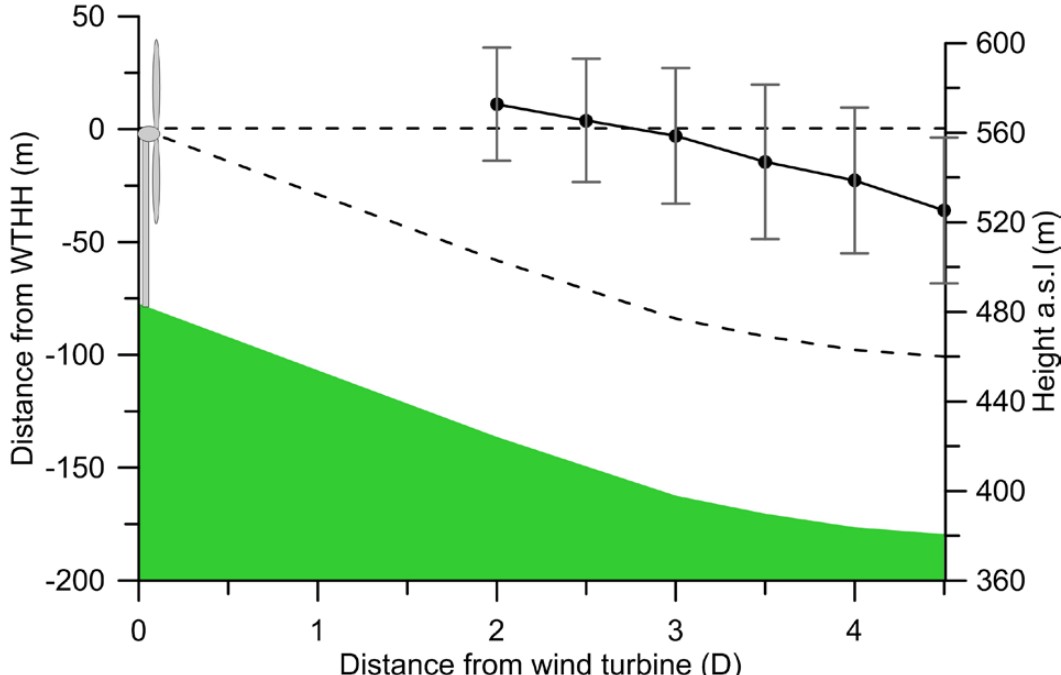

**Figure 14. The mean height of the wake centre (± one standard deviation) for distances downwind of the wind turbine at 0.5 *D* intervals from 2 to 4.5 *D* for A cases only. The terrain is shown in green. The dashed lines indicate the expected wake centre if it remained at the *WTHH* and for the case where the centre is purely terrain-following.**

The vertical location of the wake centre exhibits strong diurnal variability (as a proxy for atmospheric stability) with lower wake centres during the day when conditions are most unstable and highest overnight when conditions are stable (indicated in Figure 15 using the median value of *z/L* from 78 m height at Tower 20 from the same date/time as the wake cases). In most hours the wake centre is initially lofted, likely directed by flow over the crest, but then descends and broadly follows the slope. At 2 *D* the mean wake centre is above *WTHH* during the night-hours but has a mean value of -8 m (relative to *WTHH*) at 08:00 UTC. For every

downstream distance, the wake centres are lower in unstable, daytime conditions than in overnight stable conditions which is in direct contrast to the case studies presented in (Menke et al., 2018a). This might reflect differences in case study versus ensemble mean statistics, or differences in the lidar scan patterns. The lidar scans presented here are designed to track the wake centre whereas the near-horizontal diamond scans in (Menke et al., 2018a) are able to track the higher level flow above the valley but do not penetrate very far into the valley (below the ridgeline). The daytime hours of 09:00 to 17:00 UTC are largely associated with

unstable conditions. During these hours of the day, the mean wake centre is slightly above the equivalent *WTHH* (+3.5 m) at a downstream distance of 2 *D* to an average of -50 m by a distance of 4.5 *D* (Figures 15 and 16). In stable conditions that prevail during 18:00 to 07:00 UTC, the mean wake centre is an average of +13.5 m from *WTHH* at 2 *D* and -33 m at 4.5 *D*. Thus, there is a tendency for the wake centre to be higher in stable than unstable conditions (Figure 16). For the most extreme case in the hour 21.00 (marked L) the average of all values for the wake centre is +27 m at 2 D downstream but then descends to -38 m by 4.5 D.

In contrast, at 8:00, an hour that is defined as near-neutral, and at 13:00, an hour that is very unstable, the wake centre drops below -40 m after 3 D downstream. The behaviour of wakes is clearly very complex but despite a large amount of scatter, there is a consistent relationship between the value of *z/L* and the wake centre height for each downstream distance with wake centres in stable conditions being higher (Figures 15 and 16).

    Naturally hour of the day is an imperfect surrogate for stability due to the complexity of the relationships between wind speed,

turbulence intensity and stability and their impact on wake generation and behaviour. The velocity deficit is large for low to




moderate inflow wind speeds (Barthelmie et al., 2013) but the expansion, coherence and meandering of wakes is driven by the external flow impacted by stability, turbulence and the downstream flow characteristics. Further although the whole data set, locally determined $L$ and $TI$ are strongly linked to hour of the day (Figure 10), there is a high degree of spatial variability in near-surface stability and estimates based on similarity theory may not be fully valid for heights relevant to the wake particularly given

variability in the larger scale (synoptic) flow and the presence of thermos-topographic flows.

Wake centre location is strongly linked to inflow wind speed as estimated from data from Tower 20 (Figure 17). Wake centres are higher above ground level under lower wind speeds and their average locations are above the *WTHH* horizontal line (initially lofted) for all but the highest wind speeds. For higher inflow wind speeds, the mean wake centre is consistently further below *WTHH*. Given that the flow is perpendicular to the ridge for wind turbine wakes to be observed, then the wind speed dependence

of wake centre position may reflect the likelihood that flow at/near the wake height is more/less likely to be fully attached to the underlying terrain (see discussion in section 1.2 (Whiteman and Doran, 1993)).

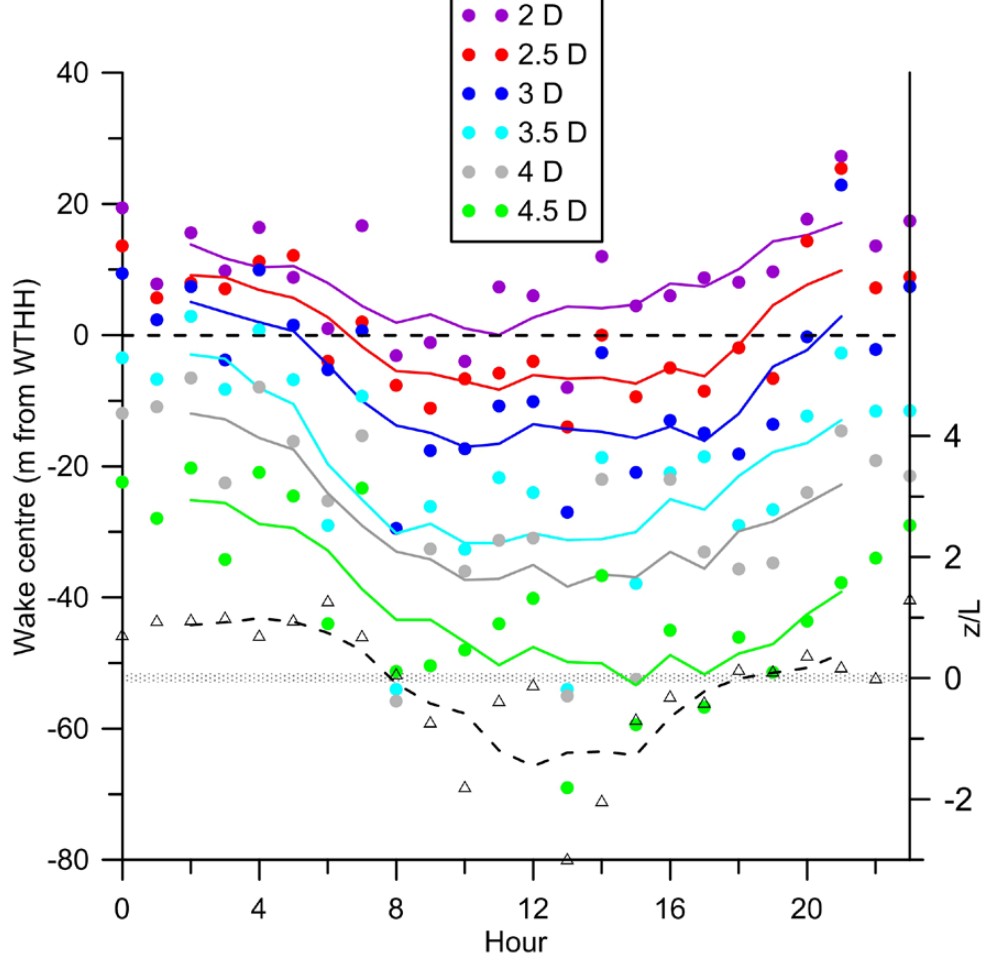

**Figure 15. Mean vertical location of the wake centre by hour of the day (A cases only) for different distances (2-4.5*D*) downstream from the wind turbine location. Coloured dots show the height of the centre relative to the wind turbine hub-height (*WTHH*- the**
**dashed line at 0 m) and the lines depict a 5-point running mean for each downstream distance. Also shown is the median value for z/L for each hour (black line/open triangles) with near-neutral conditions indicated by the shaded bar (right hand axis).**





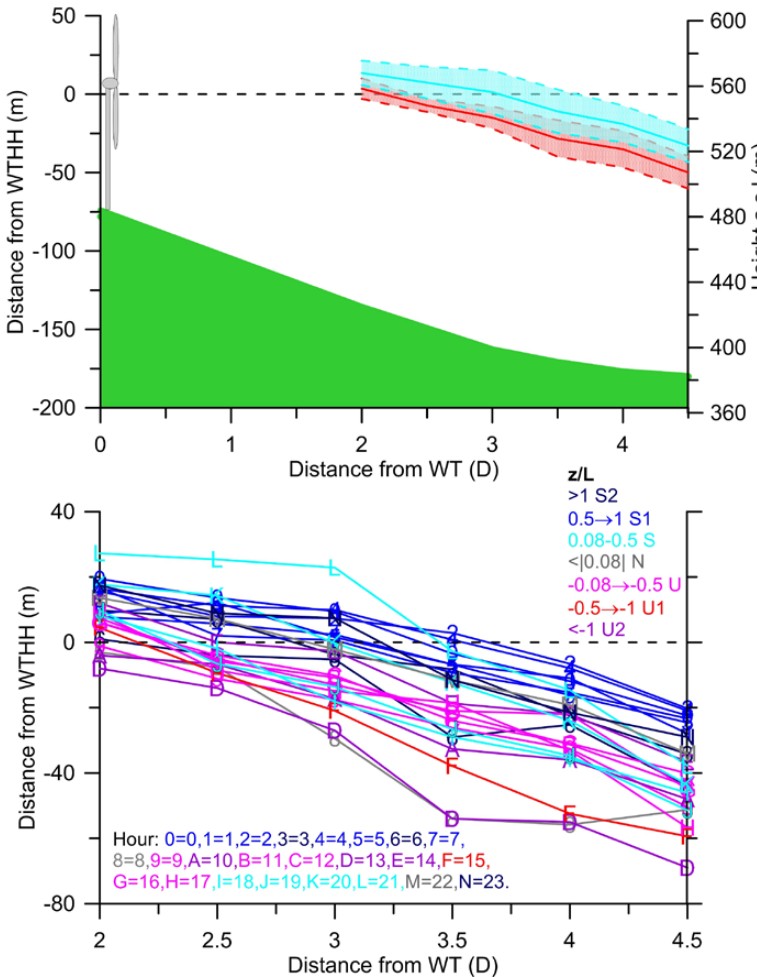

**Figure 16. Top: mean vertical location (±one standard deviation) of the wake centre by hour of the day for different distances downstream of the wind turbine location (A cases only) during hours with unstable conditions (red, median value of *z/L*<-0.08) and stable conditions (cyan, median *z/L*>0.08). Bottom: Same data as in the upper panel but shown by hour of the day. Wake centre**
5 **locations for each hour of the day (indicated by the number/letter in the inset box). Red/pink colours indicate hours typified by unstable conditions, blue colours depict hours of the day when stable conditions prevail and the grey lines depicted data from hours where conditions are frequently near-neutral.**



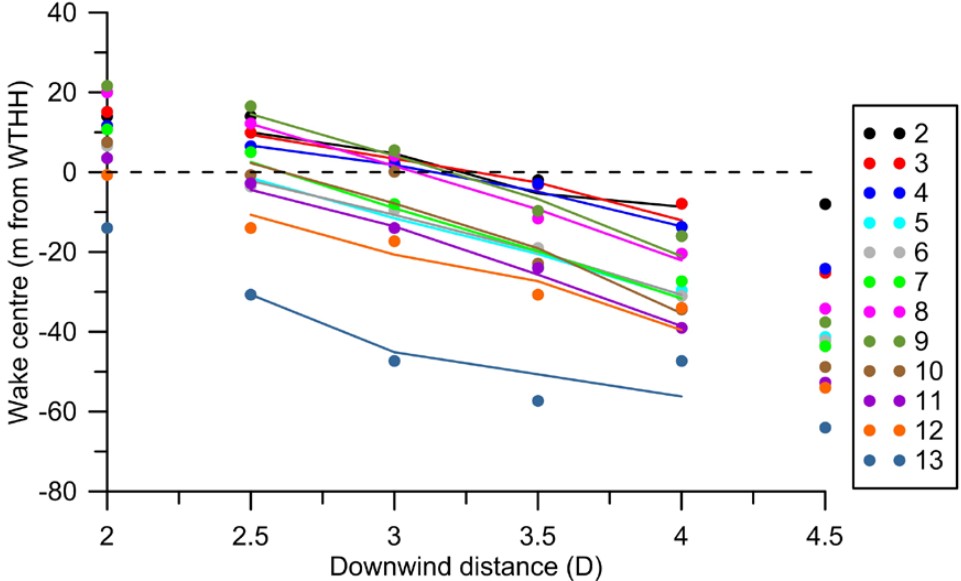

**Figure 17. Mean vertical location of the wake centre conditionally sampled by inflow tower wind speed at 78 m height (in ms⁻¹) for different distances downstream of the wind turbine location (A cases only).**

## 5 Conclusions

The behaviour and characteristics of wind turbine wakes in complex terrain are investigated using long-term measurements from scanning Doppler lidar from the Perdigão experiment in Portugal. Local meteorology (stability, wind speed, wind direction and turbulence intensity) is defined using data from 78 m (the wind turbine hub-height) on the meteorological tower close to the wind turbine. This follows a typical diurnal cycle with the predominance of stable, low turbulence conditions overnight and unstable, high turbulence conditions during the day. There are very few near-neutral conditions and the lack of a clear diurnal signal in wind

speed is indicative of multiple scale impacts on the flow, as well as physical and thermal forcing of flow.

A single wind turbine is located on a hill-crest in the double-ridge valley where height and length scales indicate the height of the inner layer is ~40-70 m and of the outer layer 230-280 m. It can be anticipated that the downstream wake flow remains within the outer-layer and is impacted also by turbulent stresses in the inner-layer (Kaimal and Finnigan, 1994). While crest-speed up is likely to be maximum at heights below the bulk of the wind turbine wake, the interaction of the hill wake and the wind turbine wake is

important.

The Doppler lidar scan geometry was designed to optimally capture data for wind turbine wakes as they move downwind of the turbine and are impacted by the flow and meteorological conditions over this complex terrain. A large data set of over 19,000 scans is obtained. An objective processing method is developed and applied identify the wake and characterise its behaviour. Here the focus is on the location of the wake centre identified by the algorithm in nearly 2,000 cases when the wind speed and direction are

conducive to wake development. The wake centre location is correctly identified relative to subjective detection in 62% of cases and missed in only 5% of cases. A clear wake centre is identified but is split into multiple lobes in a further 16% of cases which require more detailed investigation and analysis. The main differences in external meteorological conditions between the coherent and split wakes is linked to higher wind speeds and turbulence and more unstable conditions in split wakes. The remaining 33% are divided into cases where no wake can be identified – possibly because the wind turbine is not operating – and cases where the

flow pattern is so complex that neither the algorithm or a subjective approach could detect a wake signature.



Despite the complexity of the flow over the study site, the automated algorithm successfully identifies wakes when they are present and based on tracking of over 500 clearly defined wakes, the wake is initially directed by the lofting of flow over the crest and that it on average remains higher in stable low wind speed conditions and descends most rapidly under conditions of high wind speeds. Wakes are always initial lofted except in a few cases with inflow wind speeds above 11 ms$^{-1}$ which is likely from wakes following the streamlines of lofted flow over the hill crest. There is a very clear diurnal cycle with wake centres moving downslope more consistently in unstable conditions during the day and remaining at greater heights during stable conditions. There is also a consistent pattern in the location of the wake height being higher in low wind speeds, when conditions tend to be more stable/unstable. Further analysis of the wake characteristics at Perdigão will be directed towards on the magnitude of the velocity deficit and the wake expansion under different meteorological conditions.

## Data availability

All data analysed herein are available for download from the New European Wind Atlas data portal (2018) hosted by the University of Porto and accessible at; http://perdigao.fe.up.pt.

## Acknowledgments

We gratefully acknowledge funding from the U.S. National Science Foundation (1565505), the U.S. Department of Energy (DE-SC0016438) and a Fulbright Fellowship to RJB. We are grateful to the scientists and technicians of the Technical University of Denmark (DTU), INEGI, University of Porto, and the National Center for Atmospheric Research for their logistical support during the Perdigão measurement campaign and to the municipality of Vila Velha de Ródão, landowners who authorized installation of scientific equipment in their properties, the residents of Vale do Cobrão, Foz do Cobrão, Alvaiade, Chão das Servas and local businesses who kindly contributed to the success of the campaign. The space for the operational centre was generously provided by Centro Sócio-Cultural e Recreativo de Alvaiade in Vila Velha de Rodão. Grateful appreciation is extended to DTU-Wind Energy for hosting SCP and RJB during their sabbatical years, particularly Dr Hans E Jørgensen, Head of Meteorology and Remote Sensing and to Danmarks National Bank for a housing fellowship.

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
