# Peer review of "Automated Wind Turbine Wake Characterization in Complex Terrain"

_Atmospheric Measurement Techniques, 2018_

## Referee Comment (RC1) · Anonymous Referee #1 · 14 Feb 2019

In this manuscript, new analyses from the Perdigao experiment are presented. Specifically, the authors analyzed lidar measurements carried out for different wind conditions and atmospheric stability conditions. A procedure for automatic detection of wake centers from lidar measurements is proposed. This work in novel and of high interest for the wind energy community. Some results are in contrast with a previous study from the Perdigao experiment (Menke et al., 2018a), which may deserve a deeper discussion.

A very comprehensive introduction on flows over complex topography is provided in Sect. 1, followed by a description of the test site. An interesting discussion about optimal design of lidar scans is presented in Sect. 3.1. However, the selection of the used scanning parameters is not clearly justified (P8, LL 35-43). In Sect. 3, the use of the narrow PPI scans to define the background flow in not clear. I guess some details

are missing in the description of the flow analysis.

This paper states that higher wake centers are observed under stable conditions and lower under convective conditions, which is the opposite of the previous results from Menke et al., 2018a. A more detailed comparison between these two works should be provided. One of the motivations for this disagreement is that the LiDAR scans performed for this work penetrate deeper into the valley and, indeed, the wake flow evolving within the inner layer should have been measured. However, the wake centers are still in the middle/outer layer, if I am not mistaken. I suggest to provide a clearer description of the results presented in these two works, their differences and motivations for this disagreement. More details are listed below.

Comments: 1. Table 1. I am not sure retrieval of all the parameters in Table 1 is described in the text. For instance, provide details how z0 is calculated.

2. P8, L 22, what is an arc scan? A PPI scan over an azimuthal range smaller than 360 degrees? Please specify.

3. IMPORTANT: The deviation in wind direction between met-tower and lidar data is a bit puzzling. Please provide more details why such big discrepancy between the two measurement techniques, which I am not sure is only a consequence of the complex topography.

4. P3, L11: equation "(X D $\pm$ 2)" might be incomplete.

5. Fig. 4. Cross-check this figure. There are few typos in this chart.

6. P24 – Figure 13: cross-check median z/L of Case A in the bottom plot (-0.22 instead if 0.22)

7. P25. These results on the height of the wake center are very interesting. In particular, it is interesting that these results are in contrast to those of Menke et al. 2018a, which are related to the same field campaign. I guess a more detailed discussion should be provided to understand possible justifications for these discrepancies.

---

## Referee Comment (RC2) · Anonymous Referee #2 · 12 Mar 2019

The study mainly focuses on detecting the height of the wake center through a research measurement campaign in a complex site, Perdigao, Portugal. The results are novel and interesting, I recommend for publication. My major and minor comments are listed below.

Major Comments: ——————————— - In your sentences ". . .the remaining 33% could not be categorized either by the algorithm or subjectively, mainly due to the complexity of the background flow." Don't you think this percentage is a bit high? I think the information about such situations should be put into a clearer definition. In which cases the categorization is failed? Is it possible to enlarge the comments on the issue?

- It is also not clear to me why the initial free-stream in the code sometimes cannot be assigned. I am guessing, these cases are within the mentioned 33%. How do you

assume that the free-stream value derived from the radial velocity of the measured line of sight direction is sometimes valid and sometimes not?

- Do you find any similarities or differences between your results/measurements and the two experimental works done within the Larsen et al., 2008 study (Dynamic wake meandering modeling, Risoe-R, No. 1607(EN)?

Minor comments: ———————— - Although figure 1 is a good work of combining many information together I would prefer the line of sight lines "joined". Current view does not give any information due to the color mixed anyway. Furthermore, maybe a zoomed-in 2D plot of a line of sight vector plot might be helpful.

- Equation 4: k sign should be as it was defined in page 3 line 11 ($\kappa$), because "k" will be used for the rate of expansion later at Page 5 Line 14.

- In figure 2 and 14, your turbine sketches are downwind turbines, but 2 MW Enercon E-82 is an upwind turbine. One can misunderstand the setup.

---

## Author Comment (AC1) · 2 Apr 2019

Reponses begin with » See also tracked changes version of the manuscript (attached).
results are in contrast with a previous study from the Perdigao experiment (Menke et al., 2018a), which may deserve a deeper discussion. A very comprehensive introduction on flows over complex topography is provided in Sect. 1, followed by a description of the test site. An interesting discussion about optimal design of lidar scans is presented in Sect. 3.1. However, the selection of the used scanning parameters is not clearly justified (P8, LL 35-43). In Sect. 3, the use of the narrow PPI scans to define the background flow in not clear. I guess some details are missing in the description of the flow analysis. This paper states that higher wake centers are observed under stable conditions and lower under convective conditions, which is the opposite of the previous results from Menke et al., 2018a. A more detailed comparison between these two works should be provided. One of the motivations for this disagreement is that the LiDAR scans performed for this work penetrate deeper into the valley and, indeed, the wake flow evolving within the inner layer should have been measured. However, the wake centers are still in the middle/outer layer, if I am not mistaken. I suggest to provide a clearer description of the results presented in these two works, their differences and motivations for this disagreement. More details are listed below.

Comments: 1. Table 1. I am not sure retrieval of all the parameters in Table 1 is described in the text. For instance, provide details how z0 is calculated. »L 23 and Table 1. The results from using two appropriate roughness lengths are given. It is made clear now these are assumed values: "For the terrain specifications of Perdigão and flow from the southwest (i.e. inflow for wind turbine and thus wakes that potentially will enter the Galion lidar scanned volume), the inner and middle layer heights of l∼50 m (equation 1) and hm∼ 284 m (equation 6) for near-neutral stability and with z0 assumed to be either 0.3 m or 0.1 m (Table 1)."

2. P8, L 22, what is an arc scan? A PPI scan over an azimuthal range smaller than 360 degrees? Please specify. » P8 L30. It is defined now as: "Most frequently used scanning patterns comprise; one or more arc scans (in each arc scan the scan elevation angle is held constant while the azimuth angle is varied i.e. it is a pseudo Plan Position

Indicator (PPI) scan in which the azimuth angle<360) information, and/or Range Height Indicator (RHI) scans (varying elevation, fixed azimuth angle) and/or Vertical Azimuth Display (VAD) (high elevation angle, 360 scan at fixed azimuth angles).

3. IMPORTANT: The deviation in wind direction between met-tower and lidar data is a bit puzzling. Please provide more details why such big discrepancy between the two measurement techniques, which I am not sure is only a consequence of the complex topography. » We checked the direction in this sector 200-250 between the data from sonic at 60 m height on tower 20 and data from the sonic at 55 m height 333 m along the ridge. The mean difference is 3.1 and the standard deviation is 14.3 indicating substantial variability. We've added to and clarified the following text: 'There is less good agreement for wind direction. There are three main reasons for the scatter beyond the spatial offset between the mast-based 'point' measurements and the use of the average direction from the arc scan (Figure 3); i) the discretization of wind directions from the lidar is a function of the scanned azimuth angles ii) there are fundamental differences in volume-average observations from lidars and sonic anemometers (Wang et al., 2015), iii) heterogeneity in flow conditions along the ridge and turning of the flow as it summits the crest of the hill (see for example Figure 4 in (Vasiljević et al., 2017)).'

4. P3, L11: equation "(X D _ 2)" might be incomplete. »P13 L11 This X was intended to represent the different distances but given that does not seem to be clear we've changed (X D $\pm$ 20 m) to ($\pm$ 20 m).

5. Fig. 4. Cross-check this figure. There are few typos in this chart. »We found one typo 'WTHHH' which was changed to 'WTHH' and we've changed PPI to arc scan for consistency.

6. P24 – Figure 13: cross-check median z/L of Case A in the bottom plot (-0.22 instead if 0.22) »Figure 13. Thanks for pointing this out. This was checked and corrected.

7. P25. These results on the height of the wake center are very interesting. In particular, it is interesting that these results are in contrast to those of Menke et al.

2018a, which are related to the same field campaign. I guess a more detailed discussion should be provided to understand possible justifications for these discrepancies. »We've expanded the comparison but the fundamental difference is given; we have tracked the wind turbine wake centre over a distance of 4.5 D while Menke et al.'s results include the hill wake tracked over 30 D distance from the wind turbine. We've added Supplementary Material to allow additional description and please see also the new Table S1. in Supplementary Information.  

Please also note the supplement to this comment:
https://www.atmos-meas-tech-discuss.net/amt-2018-461/amt-2018-461-AC1-supplement.pdf

**Supplement:**

[revised manuscript text omitted]

A previous study analysed a more limited sample of data from scanning Doppler lidars operated from the northeast ridge during from thea 2015 campaign at the Perdigão site and sought to examine the behaviour of wakes from the wind turbine (Menke et al., 30 2018). While our statistical assessment of 1120 wake cases from 2017 show that the mean wake height is lower as it moves downwind from the crest in unstable conditions than in stable conditions, that previous study of 21 cases suggest that in unstable conditions the wakes are elevated, that they propagate horizontally in near-neutral or stable conditions and can be downwards if gravity waves are present (Menke et al., 2018). There are several differences in the research methodologies that can account for this apparent discrepancy. These are summarized in Table S1 and include:

35   1)  The studies are not for the same period with the same wind speed conditions. We have shown that in virtually all cases with an inflow wind speed $< 12 ms^{-1}$, the wake centre is initially lofted (i.e. at a height greater than the wind turbine hub-height centreline) and that by about 3 $D$ downstream most wakes start to descend. In low wind speeds (as shown for the unstable case in wakes are initially lofted and remain above the horizontal line of the wind turbine hub-height further

downstream. Of the 21 cases presented in (Menke et al., 2018) 14 showed upwards movement while 6 were propagated horizontally and 2 downwards (no information was provided regarding at which distances downstream).

2) The location and configuration of the Doppler lidars is fundamentally different. The earlier study analysed data from lidars scanning in the near horizontal (i.e. at considerable height above the intervening valley and ridge slopes). In this paper data are reported from a scanning Doppler lidar configured to scan upslope towards the wind turbine. Thus, the ability to track the wind turbine wakes differs. Use of the scanning lidar in the diamond pattern allows a horizontal view of the wake for about 4-5$D$ but does not permit wake tracking if it move substantially from the horizontal plane. Further, the flow field detected reflects a combination of the wake with the larger scale flow such as the hill wake which propagates over much longer distances than a wind turbine wake. Conversely, the lidar configuration used in this work relies principally on arc scans as positive displacements from the horizontal and permits tracking for a wider variety of wake centre locations but does not permit horizontal tracking of the wake beyond about 4.5 $D$.

3) There are major differences in the stability classification from the WRF model at 3 km resolution and in that from in situ sonic anemometer measurements.

Table S1. Comparison of measurements and results from this study and those in (Menke et al., 2018)

| | This study | Menke et al. 2018 |
|---|---|---|
| Stability measurements | $z/L$ from hub-height sonic anemometer at ridge top | Stability defined using 3 km resolution simulations with the Weather Research and Forecasting mode. eight not given |
| Stability definition | stable $L$<1000 m, neutral $L\pm$|1000 m|, unstable $L$>-1000 m | stable $1/L$ > 0.01, neutral 0.01 > $1/L$ > -0.01, unstable $1/L$< -0.01 |
| Data period | 27 Jan – 30 Jun 2017 | 8 May- 26 Jun 2015 |
| 10-minute periods analyzed | 1120 | 21 |
| Scan type | rc scans at multiple elevation angle | ear-horizontal diamond & transect scans |
| Distance from WT over which wakes are tracked | 2-4.5 $D$ | 1-~30 $D$ |

---

## Author Comment (AC2) · 2 Apr 2019

Responses begin with » See also tracked changes version of the manuscript (attached)
ing 33% could not be categorized either by the algorithm or subjectively, mainly due to the complexity of the background flow." Don't you think this percentage is a bit high? I think the information about such situations should be put into a clearer definition. In which cases the categorization is failed? Is it possible to enlarge the comments on the issue? »P22.L15. Yes. We improved the wording on this. The new wording reads: 'There are several types of D case. The most common is that it not possible to distinguish a centre of velocity deficit from the complexity of the background flow (Type D, Figure 12d), sometimes because what could potentially be the wake is split. However, in most of these cases there are other areas of much lower velocity present in the scan. The example of a case D type shown in Figure 12d is very typical of the flow complexity with weak upslope/downslope flow to the right/left of the centreline to the wind turbine (shown as lateral distance =0). This flow pattern persisted for many consecutive time periods and thus appears to represent micro-scale topographic forcing of the flow (see slope variability in Figure 1). Naturally, not all case D wake types are reflective of flow complexity. There are also a few cases where the velocity deficit is not present, use of SCADA data might remove some of these cases as it is possible that the wind turbine was not operating during all 10-minute periods.' - It is also not clear to me why the initial free-stream in the code sometimes cannot be assigned. I am guessing, these cases are within the mentioned 33%. How do you assume that the free-stream value derived from the radial velocity of the measured line of sight direction is sometimes valid and sometimes not?

»We've added some wording on this to explain it more clearly. In essence, there are some cases where there is not a strong enough signal to return a wind speed at distances > 1 km. »p11. l28. ' However, for some periods with low clouds/rain there are insufficient returned wind speeds at this distance ($\sim$1 km) to proceed.'

- Do you find any similarities or differences between your results/measurements and the two experimental works done within the Larsen et al., 2008 study (Dynamic wake meandering modeling, Risoe-R, No. 1607(EN)? » There is not much similarity because

we're focusing on the location of the wake center in the vertical plane whereas Larsen et el in this study focus more on the lateral movement. Our plan is next to analysis the horizontal/lateral movement so thanks for the reference. It would be really interesting to test out the wake meandering model on this dataset.

Minor comments: ———————— - Although figure 1 is a good work of combining many information together I would prefer the line of sight lines "joined". Current view does not give any information due to the color mixed anyway. Furthermore, maybe a zoomed-in 2D plot of a line of sight vector plot might be helpful. »We've added a scan view plot of the 15 elevation scan to show the scan pattern more clearly but the points are important because they show the location of each range gate.

- Equation 4: k sign should be as it was defined in page 3 line 11 (_), because "k" will be used for the rate of expansion later at Page 5 Line 14. »Thanks for pointing that out it has been changed in Equation 4.

- In figure 2 and 14, your turbine sketches are downwind turbines, but 2 MW Enercon E-82 is an upwind turbine. One can misunderstand the setup. »Thanks for pointing that out it has been changed on Figure 2 and 14. We've also changed it on Figure 16.

Please also note the supplement to this comment:
https://www.atmos-meas-tech-discuss.net/amt-2018-461/amt-2018-461-AC2-supplement.pdf

[Figure]

**Supplement:**

[revised manuscript text omitted]

A previous study analysed a more limited sample of data from scanning Doppler lidars operated from the northeast ridge during from thea 2015 campaign at the Perdigão site and sought to examine the behaviour of wakes from the wind turbine (Menke et al., 30 2018). While our statistical assessment of 1120 wake cases from 2017 show that the mean wake height is lower as it moves downwind from the crest in unstable conditions than in stable conditions, that previous study of 21 cases suggest that in unstable conditions the wakes are elevated, that they propagate horizontally in near-neutral or stable conditions and can be downwards if gravity waves are present (Menke et al., 2018). There are several differences in the research methodologies that can account for this apparent discrepancy. These are summarized in Table S1 and include:

35   1)  The studies are not for the same period with the same wind speed conditions. We have shown that in virtually all cases with an inflow wind speed $< 12 ms^{-1}$, the wake centre is initially lofted (i.e. at a height greater than the wind turbine hub-height centreline) and that by about 3 $D$ downstream most wakes start to descend. In low wind speeds (as shown for the unstable case in wakes are initially lofted and remain above the horizontal line of the wind turbine hub-height further

downstream. Of the 21 cases presented in (Menke et al., 2018) 14 showed upwards movement while 6 were propagated horizontally and 2 downwards (no information was provided regarding at which distances downstream).

2) The location and configuration of the Doppler lidars is fundamentally different. The earlier study analysed data from lidars scanning in the near horizontal (i.e. at considerable height above the intervening valley and ridge slopes). In this paper data are reported from a scanning Doppler lidar configured to scan upslope towards the wind turbine. Thus, the ability to track the wind turbine wakes differs. Use of the scanning lidar in the diamond pattern allows a horizontal view of the wake for about 4-5$D$ but does not permit wake tracking if it move substantially from the horizontal plane. Further, the flow field detected reflects a combination of the wake with the larger scale flow such as the hill wake which propagates over much longer distances than a wind turbine wake. Conversely, the lidar configuration used in this work relies principally on arc scans as positive displacements from the horizontal and permits tracking for a wider variety of wake centre locations but does not permit horizontal tracking of the wake beyond about 4.5 $D$.

3) There are major differences in the stability classification from the WRF model at 3 km resolution and in that from in situ sonic anemometer measurements.

Table S1. Comparison of measurements and results from this study and those in (Menke et al., 2018)

| | This study | Menke et al. 2018 |
|---|---|---|
| Stability measurements | $z/L$ from hub-height sonic anemometer at ridge top | Stability defined using 3 km resolution simulations with the Weather Research and Forecasting mode. eight not given |
| Stability definition | stable $L$<1000 m, neutral $L\pm$|1000 m|, unstable $L$>-1000 m | stable $1/L$ > 0.01, neutral 0.01 > $1/L$ > -0.01, unstable $1/L$< -0.01 |
| Data period | 27 Jan – 30 Jun 2017 | 8 May- 26 Jun 2015 |
| 10-minute periods analyzed | 1120 | 21 |
| Scan type | rc scans at multiple elevation angle | ear-horizontal diamond & transect scans |
| Distance from WT over which wakes are tracked | 2-4.5 $D$ | 1-~30 $D$ |

---

## Referee Comment (RC3) · Anonymous Referee #3 · 17 Apr 2019

Summary: The manuscript presents a method for detecting the location of the wake center from ground-based scanning Doppler lidar measurements, which is then applied to detect the wake of a wind turbine on a crest. Results show that the detection of the wake works in the majority of the cases (against a subjective control). Further, the results show that, after an initial raise in the near wake region, the wake center descends following the terrain slope with a height a. g. l. depending on the stability parameter.

Positive are the description of the challenges of scanning a wind turbine wake with a movable-head Doppler lidar and the interesting findings on the wake center evolution (especially considering the general sparsity in literature). Weak points are the description of the wake center detection itself and the presentation of some results could be

improved (see main comments below). If those issues are addressed, the paper could be considered for publication.

Main comments:

1) The description of method of the wake center detection should be improved (see detailed comments for pages 10 and 13 and Fig. 4).

2) The paper should touch upon possible false-positive detections of the wake center (see detailed comment on Fig. 12).

3) The presentation of some results could be improved and the observed dependencies could be quantified (see detailed comment on the lower panel of Fig. 16 and Fig. 17).

Language: I noticed a few missing comma and typos (see detailed comments for the ones I noticed). Given that I am not a native English speaker, there are probably more than those. Some phrasings are complicated and required me to read sentences twice to understand them.

Detail comments:

Abstract, line 10: insert "a" in "by scanning" (the information that it is a ground-based scanning Doppler lidar and that Perdigão is in Portugal could be included, too).

Abstract, line 11: "possible wake cases" could be more precise so that it relates to the wind speed and wind direction criteria.

Abstract, line 12: The first association with "spit centers" is for me the (idealized) double-peak/donut structure of the wake near the nacelle. I believe the meaning here is more general and should be described more precise.

Abstract, line 16: In connection with the comments to Fig. 16 and 17, the word "strongly" could be replaced with a quantitative statement.

Page 1, line 28: remove inner brackets at the citation

Section 1.2: It would be a great improvement, if the flow behaviors introduced in this section would be picked up by the discussion of the wake center location in section 4.4.

Page 3, line 3: "in" instead of "is"

Page 3, line 4: not sure, but I believe it is "interaction with" instead of "interaction of"

Page 3, line 5 and line 9: L_h and H are introduced twice.

Page 3, line 20: Based on Eq. (4), it seems that an overbar indicates averaging. Then the sensible heat flux in the text should also have an overbar.

Page 4, line 12-13: The text in the brackets seems to be redundant to Eq. (8).

Page 5, Eq. (11): The variable x is sometimes an uppercase letter and sometimes a lowercase letter. Are they the same? If they are the same, then its usage should be consistent.

Page 5, line 14: It might be better to start with the dependency of k on various ambient parameters and then introduce the assumption of k=0.075 to get some approximate figures.

Page 9, line 39: Is this azimuth range (199-295°) the same for all elevation angles and other heights have more/less points (i.e. azimuth resolution changes) or is the azimuth range also different at other elevation angles?

Page 10, line 20: I believe "wakes" should be singular.

Page 10, lines 23-25: I did not understand this refinement of the first estimation. Why is it done for wind directions of exactly 210° or 240°? As I understood it, this refinement works similar to first estimate, but with an extended vertical range and more elevation angles. Is that correct? (The sentence could be rephrased for better understanding).

Fig. 4: A lot of the boxes have unclear labels and in some cases I cannot understand what the algorithm is doing exactly. I believe some of my problems come from an

inconsistent use of variables/names. Below are the problems I have:

a) Differentiation of VAD DIR (wind direction retrieved from VAD scan?), PPI DIR (wind direction from the arc scans?), wake direction (the direction from the wind turbine to the Doppler lidar?), DIR (no idea), and wake DIR (short for wake direction?)

b) VAD DIR is used in the third check, but later it is checked, whether VAD values can be retrieved.

c) What is checked at the box with the label "Inflow U/DIR"?

d) WTHHH should be WTHH?

e) Wake center location has arrows coming in from both "Vertical slice 2-6D" (I assume this would be the un-interpolated data) and "Cubic spline interpolation" (the interpolated data). According to the text, it is only detected from the interpolated data. If not, how are they combined?

f) I assume that the box with "Gaussian fit" and "Wake metrics..." is part of the mentioned future work and not relevant for wake center location analyzed in this paper? Then they could greyed-out or removed.

Page 13, line 7-8: The velocity deficit is introduced with two variables (delta U and v_d).

Page 13, line 11: What does the "X D +/- 20 m" in the brackets mean?

Page 13, line 13-14: The sentence "Then the plane of radial velocities is discretized into 20 m horizontal planes and a mean radial velocity is computed." needs some clarifications: a) Are the radial velocities averaged or a corrected velocity depending on az/el of the beams and wind direction? b) If I understood correctly, the 20 m are referring to the vertical height of each plane – so that in the end a U_0 is gained that only depends on y and z?

Page 13, line 15-16: The detection of the wake center should be explained in more detail. As I understood it, the algorithm searches the nearest local maximum from x=z=0. I did not understand what is meant with "refining that location by checking it moving around the grid cells". From the next paragraph, I understand that an interpolation takes place before detection of the wake center - that should be mentioned here.

Fig. 5: It should be stated that axis are distance from the lidar.

Fig. 6 and Fig. 7: x and z could be used for axis labels.

Page 16, line 13: comma before but

Section 4.2 (headline): I believe "Data availability" would better describe the content of this section.

Page 22, line 4: I do not understand what is meant with "wake centre of gravity".

Page 22, line 12: "at" instead of "as".

Page 22, line 14: comma before but

Page 22, line 14: I believe it should be either "identifies a part of the wake" or "identifies parts of the wake".

Page 26, line 11: Remove inner brackets and replace with "or" or "; also see" (I first thought it is a reference to section 1.2 in Whiteman and Doran).

Fig. 12: The example of wake type B (and comparing it with with type D) makes me concerned about misdetections. How does the wake center of type B cases develop downwind – i.e does it evolve in a continuous manner or is the wake center "jumping arround" for successive downwind distances? For an automated wake detection, a quality flag system providing indication on the reliability of the detected wakes would be a great enhancement (beside the above, other possible routes could be the spatial standard deviation compared with the amplitude of the detected wake center or the number of local minima/maxima above a certain threshold).

Fig. 16, lower panel: The usage of the symbols to show the hour of day is not working, because they cannot be read within the figure. And I could not make sense of the text in the top right (I got the z/L intervals, but what is the meaning of the S, N and U?). In my opinion, the time dependency is better illustrated in Fig. 15 and therefore Fig. 16 should focus on the dependency with the stability parameter. One idea could be, to pick one exemplary downwind distance and plot distance from WTHH vs. z/L. Then a linear fit /correlation (if significant) could quantify the relation with stability parameter. The remaining downwind distances could be reported in the text or in a table.

Fig. 17: The color coding is unfortunate, because two similar greens and purples are used. Grey scales or a linear color map (e.g. blue -> red) would better illustrate a dependency. Similar to the comment on Fig. 16, the dependency could be quantified with a linear fit and correlation (if significant).

Section 5 (headline): "Summary" would be a better description for the content of the section.

Page 28, line 10: What is physical forcing?

Page 28, line 18: Insert "to" between "applied identify"

---

## Author Comment (AC3) · 25 Apr 2019

This is a response to Reviewer 3: Interactive comment on "Automated Wind Turbine Wake Characterization in Complex Terrain" by Rebecca J. Barthelmie and Sara C. Pryor Atmos. Meas. Tech. Discuss., doi:10.5194/amt-2018-461-RC3, 2019

Our responses are given with ». A tracked changes version of the paper is attached. NB. The figures are shown at very low resolution to minimize the file size but they will be uploaded at high resolution for the paper.

Anonymous Referee #3 Summary: The manuscript presents a method for detecting the location of the wake center from ground-based scanning Doppler lidar measurements, which is then applied to detect

the wake of a wind turbine on a crest. Results show that the detection of the wake works in the majority of the cases (against a subjective control). Further, the results show that, after an initial raise in the near wake region, the wake center descends following the terrain slope with a height a. g. l. depending on the stability parameter. Positive are the description of the challenges of scanning a wind turbine wake with a movable-head Doppler lidar and the interesting findings on the wake center evolution (especially considering the general sparsity in literature). Weak points are the description of the wake center detection itself and the presentation of some results could be improved (see main comments below). If those issues are addressed, the paper could be considered for publication. Main comments: 1) The description of method of the wake center detection should be improved (see detailed comments for pages 10 and 13 and Fig. 4). 2) The paper should touch upon possible false-positive detections of the wake center (see detailed comment on Fig. 12). 3) The presentation of some results could be improved and the observed dependencies could be quantified (see detailed comment on the lower panel of Fig. 16 and Fig. 17).

Language: I noticed a few missing comma and typos (see detailed comments for the ones I noticed). Given that I am not a native English speaker, there are probably more than those. Some phrasings are complicated and required me to read sentences twice to understand them. »We have clarified the text. Please see tracked changes version and the detailed explanation below.

Detail comments: Abstract, line 10: insert "a" in "by scanning" (the information that it is a ground-based scanning Doppler lidar and that Perdigão is in Portugal could be included, too). »Done. This now reads: 'An automated wind turbine wake characterization algorithm has been developed and applied to a dataset of over 19,000 scans measured by a ground-based scanning Doppler lidar at Perdigão, Portugal over the period January to June 2017.'

Abstract, line 11: "possible wake cases" could be more precise so that it relates to the wind speed and wind direction criteria. Added: "Potential wake cases are identified by

wind direction and the availability of retrieved wind speeds for both the freestream and the potential wake locations up to 4.5 rotor diameters from the wind turbine"

Abstract, line 12: The first association with "spit centers" is for me the (idealized) double-peak/donut structure of the wake near the nacelle. I believe the meaning here is more general and should be described more precise. »Changed to: 'The algorithm correctly identifies the wake centre position in 62% of possible wake cases, 46% having a clear and well-defined wake centre surrounded by a coherent area of lower wind speeds while 16% have split centres or multiple lobes where the lower wind speed volumes are no longer in coherent areas but present as two or more distinct areas or lobes.'

Abstract, line 16: In connection with the comments to Fig. 16 and 17, the word "strongly" could be replaced with a quantitative statement. »This has been modified to read: 'However, this behaviour is strongly linked to hour of the day and atmospheric stability. Overnight and in stable conditions, the average height of the wake centre is 10 m higher than in unstable conditions at 2 D downstream from the wind turbine and 17 m higher at 4.5 D downstream.'

Page 1, line 28: remove inner brackets at the citation »Done

Section 1.2: It would be a great improvement, if the flow behaviors introduced in this section would be picked up by the discussion of the wake center location in section 4.4. » We have modified Figure 14 to link back to the discussion of the inner-layer height and also introduced new text: 'Although the standard deviation of the wake centre height at each downstream distance is large, the tendency of the wake centre to initially loft and then move down the slope, broadly following the grade of the terrain, is clear. It is worth noting that the wake also expands as it moves downstream. Using equation (11), the wake width will be expanded from 82 m to 107 m after 2 D and to 137 m after 4.5 D (Figure 14). Although the tendency is for the whole wake to remain above the inner layer (discussed in Section 1.2), the lower edge of the wake is within 12 m of the

inner-layer height (and the uncertainty on both heights means it is plausible that the wake volume interacts with the inner layer, especially in unstable conditions. ' (Revised Figure 14 in attached file) Page 3, line 3: "in" instead of "is" »Done

Page 3, line 4: not sure, but I believe it is "interaction with" instead of "interaction of" »Done

Page 3, line 5 and line 9: L_h and H are introduced twice. »Changed (on what is now line 18)

Page 3, line 20: Based on Eq. (4), it seems that an overbar indicates averaging. Then the sensible heat flux in the text should also have an overbar. »Done

Page 4, line 12-13: The text in the brackets seems to be redundant to Eq. (8). » We've clarified this.

Page 5, Eq. (11): The variable x is sometimes an uppercase letter and sometimes a lowercase letter. Are they the same? If they are the same, then its usage should be consistent. » Lowercase x is used to indicate a location, like a co-ordinate whereas uppercase X is used to define a distance downstream of the wind turbine.

Page 5, line 14: It might be better to start with the dependency of k on various ambient parameters and then introduce the assumption of k=0.075 to get some approximate figures. »This value of k is recommended for use in the Jensen WAsP model for land sites. We've clarified that. 'where $D_w$ is wake width in rotor diameters (D), $D_0$ is the initial wake width, k is the rate of expansion (0.075 is recommended in WAsP for land sites (Katic et al., 1986), (Troen and Petersen, 1989)), which is determined by the factors listed above such as ambient turbulence intensity and X is the distance downstream.'

Page 9, line 39: Is this azimuth range (199-295_) the same for all elevation angles and other heights have more/less points (i.e. azimuth resolution changes) or is the azimuth range also different at other elevation angles? » This has been clarified –

also in Revised Figure 1 we now show the azimuth angles in a plan view inset figure. "Elevation angles are; 9, 10.5, 12, 13.5, 15, 16.5, 18, 19.5, 21 and 23ïĆř. The arc angle in the azimuth varies with the elevation angle. For 10.5ïĆř elevation and above, it comprises 23 beams from 199-259ïĆř with resolution between 1.5 and 6ïĆř (Figure 1). For 9ïĆř elevation, some of the outermost angles were removed because they returned very little data once vegetation had grown."

Page 10, line 20: I believe "wakes" should be singular. » Done

Page 10, lines 23-25: I did not understand this refinement of the first estimation. Why is it done for wind directions of exactly 210_ or 240_? As I understood it, this refinement works similar to first estimate, but with an extended vertical range and more elevation angles. Is that correct? (The sentence could be rephrased for better understanding). »p10, l31 We have rewritten this section to clarify the process: 'In the analyses presented herein the process for identifying potential wind turbine wakes cases that may have been sampled by the Galion lidar is multi-step. The VAD scans are used for the initial screening because they are much faster to process and determine whether there can potentially be a measured wake, depending on the wind speed and direction. First, the approximate wind speed and direction at a height above the Galion lidar equal to the WTHH ïĆś 30 m are estimated from the VAD scans as the maximum negative mean value of all valid radial velocities (where a signal to noise threshold (SNR) of 1.015 is applied) within each 10-minute period. If this analysis also indicated a wind direction of 210 or 240ïĆř (i.e. flow from the wind turbine toward the scanning lidar), the processing continues using the larger volume of the arc scans from the same 10-minute period. Arc scan radial velocities (SNR > 1.015) at a range of the distance to the WTHH+40 m and for scans at an elevation angle of 12-17ïĆř are used to refine the estimated inflow wind direction and wind speed. However, for some periods with low clouds/rain there are insufficient returned wind speeds at this distance (∼1 km) to proceed.'

Fig. 4: A lot of the boxes have unclear labels and in some cases I cannot understand what the algorithm is doing exactly. I believe some of my problems come from inconsistent use of variables/names. Below are the problems I have: a) Differentiation of VAD DIR (wind direction retrieved from VAD scan?), PPI DIR (wind direction from the arc scans?), wake direction (the direction from the wind turbine to the Doppler lidar?), DIR (no idea), and wake DIR (short for wake direction?) »These are now defined on p6 and we use the same nomenclature throughout the paper and in this Figure: p6. 'Most frequently used scanning patterns comprise; one or more arc scans (in each arc scan the scan elevation angle is held constant while the azimuth angle is varied i.e. it is a pseudo Plan Position Indicator (PPI) scan in which the azimuth angle<360ïĆř) information, and/or Range Height Indicator (RHI) scans (varying elevation, fixed azimuth angle) and/or Vertical Azimuth Display (VAD) (high elevation angle, 360ïĆř scan at fixed azimuth angles).' Figure 4 caption: 'The following abbreviations are used DIR=direction, U=wind speed, VAD DIR=direction retrieved from VAD scans, arc scan DIR= direction retrieved from arc scans.' b) VAD DIR is used in the third check, but later it is checked, whether VAD values can be retrieved. »This has been changed to 'Retrievable VAD U' c) What is checked at the box with the label "Inflow U/DIR"? »We have changed this to' background U/DIR' and it is now consistent with the descriptive text on p12, l8: 'Then for each of the downstream distances considered (i.e. the vertical planes located at 2 D, 2.5 D, 3 D, 3.5 D, 4 D, 4.5 D) an assessment is made of whether there are sufficient retrieved radial wind speeds (i.e. measurements with a SNR > 1.015) to describe both the presence of a wake and the background flow field for each downstream distance and height.'

d) WTHHH should be WTHH? » This typo was corrected

e) Wake center location has arrows coming in from both "Vertical slice 2-6D" (I assume this would be the un-interpolated data) and "Cubic spline interpolation" (the interpolated data). According to the text, it is only detected from the interpolated data. If not, how are they combined? »This text has been clarified p12. l10: 'Once each case has passed these filters, radial wind fields on these planes are used to derive anomaly fields (see section 3.4) from which the wake centre location is identified and other wake

metrics are calculated for each downstream distance. The wake centre is calculated using both original data and cubic-spline interpolated fields.'

f) I assume that the box with "Gaussian fit" and "Wake metrics: : :" is part of the mentioned future work and not relevant for wake center location analyzed in this paper? Then they could greyed-out or removed. »The Gaussian fit was greyed out. As indicated, wake metrics include the wake centre location.

Page 13, line 7-8: The velocity deficit is introduced with two variables (delta U and v_d). »This has been corrected

Page 13, line 11: What does the "X D +/- 20 m" in the brackets mean? » This was changed in response to the comments of a previous reviewer.

Page 13, line 13-14: The sentence "Then the plane of radial velocities is discretized into 20 m horizontal planes and a mean radial velocity is computed." needs some clarifications: a) Are the radial velocities averaged or a corrected velocity depending on az/el of the beams and wind direction? b) If I understood correctly, the 20 m are referring to the vertical height of each plane – so that in the end a $U\_0$ is gained that only depends on y and z? »Yes. That is correct. A background radial wind speed is determined for each distance downstream at discrete heights of 20 m. This has been clarified p14, l16 and we also refer to Figure 6. 'At each of these downstream distances ($\pm$ 20 m) the radial wind speeds at each x location (lateral displacement distance from a direct transect to the wind turbine) and z height (where z is defined from the elevation of the wind turbine hub-height) are retrieved for each 10-minute period. Then the vertical plane of radial velocities is discretized into 20 m horizontal planes and a mean radial velocity is computed for each 20m plane (see Figure 6).'

Page 13, line 15-16: The detection of the wake center should be explained in more detail. As I understood it, the algorithm searches the nearest local maximum from x=z=0.I did not understand what is meant with "refining that location by checking it moving around the grid cells". From the next paragraph, I understand that an inter-
polation takes place before detection of the wake center - that should be mentioned here. »This has been clarified p14, l19: 'Anomalies from that 'background' profile are then interpolated using cubic spline interpolation and used in the wake centreline identification. The location of the wake centre is determined using the maximum velocity deficit anomaly for each height starting its search at the expected location (WTHH). The location is refined by moving from that location horizontally replacing the wake centre if the new grid cell velocity deficit is greater than the previous maximum velocity deficit value. Once locations have been checked in each 20 m horizontal plane, the algorithm moves to the next vertical plane and checks that searching for the maximum velocity deficit value. The algorithm assumes the wake has moved further downstream than the immediate rotor plane after the double bell wake shape is expanded into a near Gaussian shape (Barthelmie et al., 2003) and hence that there is a well-defined centre. ' Fig. 5: It should be stated that axis are distance from the lidar. »This is in the caption to Figure 5. 'Each panel is 1000 m by 1000 m with the Galion located at (0,0) and depicts line of sight (radial) wind speeds from each elevation angle; 7, 9, 10.5, 12, 13.5, 15, 16.5, 18, 19.5, 21 and 23ïĆř.'

Fig. 6 and Fig. 7: x and z could be used for axis labels. » The current version that states explicitly 'Height from WTHH' and 'Lateral distance' is clearer

Page 16, line 13: comma before but » A comma is not needed here. Commas are not required before the word 'but' unless two independent clauses are joined by the conjunction

Section 4.2 (headline): I believe "Data availability" would better describe the content of this section. Changed to '4.2 Data processing methodology and data availability'

Page 22, line 4: I do not understand what is meant with "wake centre of gravity". This was changed to p23, l3' B) type where the wake centre is split but the wake centre is broadly identified by the algorithm,'

Page 22, line 12: "at" instead of "as". » Done

Page 22, line 14: comma before but » Commas are not required before the word 'but' unless two independent clauses are joined by the conjunction

Page 22, line 14: I believe it should be either "identifies a part of the wake" or "identifies parts of the wake". » Done.

Page 26, line 11: Remove inner brackets and replace with "or" or "; also see" (I first thought it is a reference to section 1.2 in Whiteman and Doran). »Done

Fig. 12: The example of wake type B (and comparing it with with type D) makes me concerned about misdetections. How does the wake center of type B cases develop downwind – i.e does it evolve in a continuous manner or is the wake center "jumping arround" for successive downwind distances? For an automated wake detection, a quality flag system providing indication on the reliability of the detected wakes would be a great enhancement (beside the above, other possible routes could be the spatial standard deviation compared with the amplitude of the detected wake center or the number of local minima/maxima above a certain threshold). » There is a lot of work to do on these type B wakes. Thanks for the interesting suggestions. Given the length of the current manuscript we decided to focus on metrics of the A type cases here and follow with a detailed investigation of the much more complex B cases in a subsequent manuscript. For your interest, the lobes seem to be coherent as they move downstream but a much more comprehensive analysis is needed to confirm this and the other detail of their characteristics and behavior.

Fig. 16, lower panel: The usage of the symbols to show the hour of day is not working, because they cannot be read within the figure. And I could not make sense of the text in the top right (I got the z/L intervals, but what is the meaning of the S, N and U?). In my opinion, the time dependency is better illustrated in Fig. 15 and therefore Fig. 16 should focus on the dependency with the stability parameter. One idea could be, to pick one exemplary downwind distance and plot distance from WTHH vs. z/L. Then a linear fit /correlation (if significant) could quantify the relation with stability parameter. The

remaining downwind distances could be reported in the text or in a table. » The main point is to show both the general tendency that is really clear in the above figure and the variability by both hour and stability class. This helps to show that while the averages are really distinct there is still variability. It is not important to be able to distinguish the numbers on each line. We've added a detailed description of the two legends and also rewritten this section to illustrate the important points made by this Figure more clearly as below. The changes are clear in the tracked version of the paper provided: 'Thus, there is a tendency for the wake centre to be higher in stable than unstable conditions (Figure 16). The daytime hours of 09:00 to 17:00 UTC are largely associated with unstable conditions. During these hours of the day, the mean wake centre is slightly above the equivalent WTHH (+3.5 m) at a downstream distance of 2 D to an average of -50 m by a distance of 4.5 D (Figures 15 and 16). In stable conditions that prevail during 18:00 to 07:00 UTC, the mean wake centre is an average of +13.5 m from WTHH at 2 D and -33 m at 4.5 D. Most stable hours have wake centre trajectories that are higher than the majority of those in unstable hours (see the groupings of stable wake centre trajectories in blue colours vs the unstable wake centre trajectories in red colours in Figure 16). Despite this clear signal, there is also variability both in the grouping of individual hours into different stability classes and the height of the wake centre trajectory by stability. For the most extreme case of lofting, in the hour 21.00 (marked L) conditions are stable and the average of all values for the wake centre is +27 m at 2 D downstream but then descends to -38 m by 4.5 D. In contrast, at 8:00, an hour that is defined as near-neutral (marked 8), and at 13:00, an hour that is very unstable (marked D), the wake centre drops below -40 m after 3 D downstream. The behaviour of wakes is clearly very complex but despite a large amount of scatter, there is a consistent relationship between the value of z/L and the wake centre height for each downstream distance with wake centres in stable conditions being higher (Figures 15 and 16). ' Fig. 17: The color coding is unfortunate, because two similar greens and purples are used. Grey scales or a linear color map (e.g. blue -> red) would better illustrate a dependency. Similar to the comment on Fig. 16, the dependency could

be quantified with a linear fit and correlation (if significant). »The point is to show the general tendency of the relationship with wind speed. We evaluated various options and came up with the use of distinct symbols which, when combined with the colours, has improved this figure.

Section 5 (headline): "Summary" would be a better description for the content of the section. Changed to 'Summary and Conclusions'

Page 28, line 10: What is physical forcing? Physical forcing is generally understood to mean forces affecting on flow like the pressure gradient force, Coriolis force acting rather than thermally-generated mechanisms. To clarify this sentence we have changed it to 'There are very few near-neutral conditions and the lack of a clear diurnal signal in wind speed is indicative of multiple scale impacts on the flow, including both physical and thermal forcing. ' Page 28, line 18: Insert "to" between "applied identify" » Done

Please also note the supplement to this comment:
https://www.atmos-meas-tech-discuss.net/amt-2018-461/amt-2018-461-AC3-supplement.pdf

**Supplement:**

**Rebecca J. Barthelmie and Sara C. Pryor**
The reviewer's comments are given in italics.
Our responses are given with >>.
A tracked changes version of the paper is attached. NB. The figures are shown at very low resolution to minimize the file size but they will be uploaded at high resolution for the paper.

*Anonymous Referee #3*

*Summary: The manuscript presents a method for detecting the location of the wake*
*center from ground-based scanning Doppler lidar measurements, which is then applied*
*to detect the wake of a wind turbine on a crest. Results show that the detection of*
*the wake works in the majority of the cases (against a subjective control). Further,*
*the results show that, after an initial raise in the near wake region, the wake center*
*descends following the terrain slope with a height a. g. l. depending on the stability*
*parameter.*
*Positive are the description of the challenges of scanning a wind turbine wake with a*
*movable-head Doppler lidar and the interesting findings on the wake center evolution*
*(especially considering the general sparsity in literature). Weak points are the description*
*of the wake center detection itself and the presentation of some results could be*
*improved (see main comments below). If those issues are addressed, the paper could*
*be considered for publication.*
*Main comments:*
*1) The description of method of the wake center detection should be improved (see*
*detailed comments for pages 10 and 13 and Fig. 4).*
*2) The paper should touch upon possible false-positive detections of the wake center*
*(see detailed comment on Fig. 12).*
*3) The presentation of some results could be improved and the observed dependencies*
*could be quantified (see detailed comment on the lower panel of Fig. 16 and Fig. 17).*

*Language: I noticed a few missing comma and typos (see detailed comments for the*
*ones I noticed). Given that I am not a native English speaker, there are probably more*
*than those. Some phrasings are complicated and required me to read sentences twice*
*to understand them.*
>>We have clarified the text. Please see tracked changes version and the detailed explanation below.

*Detail comments:*
*Abstract, line 10: insert "a" in "by scanning" (the information that it is a ground-based*
*scanning Doppler lidar and that Perdigão is in Portugal could be included, too).*
>>Done.
This now reads:
'An automated wind turbine wake characterization algorithm has been developed and applied to a dataset of over 19,000 scans measured by a ground-based scanning Doppler lidar at Perdigão, Portugal over the period January to June 2017.'

*Abstract, line 11: "possible wake cases" could be more precise so that it relates to the*
*wind speed and wind direction criteria.*
Added: "Potential wake cases are identified by wind direction and the availability of retrieved wind speeds for both the freestream and the potential wake locations up to 4.5 rotor diameters from the wind turbine"

*Abstract, line 12: The first association with "spit centers" is for me the (idealized)*
*double-peak/donut structure of the wake near the nacelle. I believe the meaning here*
*is more general and should be described more precise.*
>>Changed to: 'The algorithm correctly identifies the wake centre position in 62% of possible wake cases, 46% having a clear and well-defined wake centre surrounded by a coherent area of lower wind speeds while 16% have split centres or multiple lobes where the lower wind speed volumes are no longer in coherent areas but present as two or more distinct areas or lobes.'

*Abstract, line 16: In connection with the comments to Fig. 16 and 17, the word*
*"strongly" could be replaced with a quantitative statement.*
>>This has been modified to read:
'However, this behaviour is strongly linked to hour of the day and atmospheric stability. Overnight and in stable conditions, the average height of the wake centre is 10 m higher than in unstable conditions at 2 *D* downstream from the wind turbine and 17 m higher at 4.5 *D* downstream.'

*Page 1, line 28: remove inner brackets at the citation*
>>Done

*Section 1.2: It would be a great improvement, if the flow behaviors introduced in this*
*section would be picked up by the discussion of the wake center location in section 4.4.*
>> We have modified Figure 14 to link back to the discussion of the inner-layer height and also introduced new text:
'Although the standard deviation of the wake centre height at each downstream distance is large, the tendency of the wake centre

to initially loft and then move down the slope, broadly following the grade of the terrain, is clear. It is worth noting that the wake

also expands as it moves downstream. Using equation (11), the wake width will be expanded from 82 m to 107 m after 2 D and to

137 m after 4.5 D (Figure 14). Although the tendency is for the whole wake to remain above the inner layer (discussed in Section

1.2), the lower edge of the wake is within 12 m of the inner-layer height (and the uncertainty on both heights means it is plausible

that the wake volume interacts with the inner layer, especially in unstable conditions. '

[Figure]

**Figure 14. The mean height of the wake centre is shown with a solid black line (± one standard deviation in solid grey lines) for distances downwind of the wind turbine at 0.5 *D* intervals from 2 to 4.5 *D* for A types only. The width of the wake indicated by Equation 11 is shown as the black dash-dot lines. The terrain is shown in green. The grey dashed lines indicate the expected wake centre if it remained at the *WTHH* and if the centre is purely terrain-following. The solid blue lines indicate the range of heights for the inner layer *l* from Table 1.**

*Page 3, line 3: "in" instead of "is"*
>>Done

*Page 3, line 4: not sure, but I believe it is "interaction with" instead of "interaction of"*
>>Done

*Page 3, line 5 and line 9: L_h and H are introduced twice.*
>>Changed (on what is now line 18)

5 *Page 3, line 20: Based on Eq. (4), it seems that an overbar indicates averaging. Then*
*the sensible heat flux in the text should also have an overbar.*
>>Done

*Page 4, line 12-13: The text in the brackets seems to be redundant to Eq. (8).*
10 >> We've clarified this.

*Page 5, Eq. (11): The variable x is sometimes an uppercase letter and sometimes a*
*lowercase letter. Are they the same? If they are the same, then its usage should be*
*consistent.*
15 >> Lowercase x is used to indicate a location, like a co-ordinate whereas uppercase X is used to define a distance downstream of
the wind turbine.

*Page 5, line 14: It might be better to start with the dependency of k on various ambient*
*parameters and then introduce the assumption of k=0.075 to get some approximate*
20 *figures.*
>>This value of k is recommended for use in the Jensen WAsP model for land sites. We've clarified that.
'where $D_w$ is wake width in rotor diameters ($D$), $D_0$ is the initial wake width, $k$ is the rate of expansion (0.075 is recommended in
WAsP for land sites (Katic et al., 1986), (Troen and Petersen, 1989)), which is determined by the factors listed above such as
ambient turbulence intensity and $X$ is the distance downstream.'
25
*Page 9, line 39: Is this azimuth range (199-295_) the same for all elevation angles and*
*other heights have more/less points (i.e. azimuth resolution changes) or is the azimuth*
*range also different at other elevation angles?*
>> This has been clarified – also in Revised Figure 1 we now show the azimuth angles in a plan view inset figure.
30 "Elevation angles are; 9, 10.5, 12, 13.5, 15, 16.5, 18, 19.5, 21 and 23°. The arc angle in the azimuth varies with the elevation angle.
For 10.5° elevation and above, it comprises 23 beams from 199-259° with resolution between 1.5 and 6° (Figure 1). For 9°
elevation, some of the outermost angles were removed because they returned very little data once vegetation had grown."

*Page 10, line 20: I believe "wakes" should be singular.*
35 >> Done

*Page 10, lines 23-25: I did not understand this refinement of the first estimation. Why*
*is it done for wind directions of exactly 210_ or 240_? As I understood it, this refinement*
*works similar to first estimate, but with an extended vertical range and more elevation*
40 *angles. Is that correct? (The sentence could be rephrased for better understanding).*
>>p10, l31 We have rewritten this section to clarify the process:
'In the analyses presented herein the process for identifying potential wind turbine wakes cases that may have been sampled by
the Galion lidar is multi-step. The VAD scans are used for the initial screening because they are much faster to process and
determine whether there can potentially be a measured wake, depending on the wind speed and direction. First, the approximate
45 wind speed and direction at a height above the Galion lidar equal to the *WTHH* ± 30 m are estimated from the VAD scans as the
maximum negative mean value of all valid radial velocities (where a signal to noise threshold (SNR) of 1.015 is applied) within
each 10-minute period. If this analysis also indicated a wind direction of 210 or 240° (i.e. flow from the wind turbine toward the
scanning lidar), the processing continues using the larger volume of the arc scans from the same 10-minute period. Arc scan radial
velocities (SNR > 1.015) at a range of the distance to the *WTHH*+40 m and for scans at an elevation angle of 12-17° are used to
50 refine the estimated inflow wind direction and wind speed. However, for some periods with low clouds/rain there are insufficient
returned wind speeds at this distance (~1 km) to proceed.'

*Fig. 4: A lot of the boxes have unclear labels and in some cases I cannot understand*
*what the algorithm is doing exactly. I believe some of my problems come from inconsistent use of variables/names. Below are the*
55 *problems I have:*

*a) Differentiation of VAD DIR (wind direction retrieved from VAD scan?), PPI DIR (wind*
*direction from the arc scans?), wake direction (the direction from the wind turbine to*
*the Doppler lidar?), DIR (no idea), and wake DIR (short for wake direction?)*
>>These are now defined on p6 and we use the same nomenclature throughout the paper and in this Figure:

p6. 'Most frequently used scanning patterns comprise; one or more arc scans (in each arc scan the scan elevation angle is held constant while the azimuth angle is varied i.e. it is a  pseudo Plan Position Indicator (PPI) scan in which the azimuth angle<360°) information, and/or Range Height Indicator (RHI) scans (varying elevation, fixed azimuth angle) and/or Vertical Azimuth Display (VAD) (high elevation angle, 360° scan at fixed azimuth angles).'

5 Figure 4 caption: 'The following abbreviations are used DIR=direction, U=wind speed, VAD DIR=direction retrieved from VAD scans, arc scan DIR= direction retrieved from arc scans.'

*b) VAD DIR is used in the third check, but later it is checked, whether VAD values can*
*be retrieved.*
>>This has been changed to 'Retrievable VAD U'

10 *c) What is checked at the box with the label "Inflow U/DIR"?*
>>We have changed this to' background U/DIR' and it is now consistent with the descriptive text on p12, l8: 'Then for each of the downstream distances considered (i.e. the vertical planes located at 2 *D*, 2.5 *D*, 3 *D*, 3.5 *D*, 4 *D*, 4.5 *D*) an assessment is made of whether there are sufficient retrieved radial wind speeds (i.e. measurements with a SNR > 1.015) to describe both the presence of a wake and the background flow field for each downstream distance and height.'

*d) WTHHH should be WTHH?*
>> This typo was corrected

*e) Wake center location has arrows coming in from both "Vertical slice 2-6D" (I assume*
20 *this would be the un-interpolated data) and "Cubic spline interpolation" (the interpolated*
*data). According to the text, it is only detected from the interpolated data. If not, how*
*are they combined?*
>>This text has been clarified p12. l10:
'Once each case has passed these filters, radial wind fields on these planes are used to derive anomaly fields (see section 3.4) from
25 which the wake centre location is identified and other wake metrics are calculated for each downstream distance. The wake centre is calculated using both original data and cubic-spline interpolated fields.'

*f) I assume that the box with "Gaussian fit" and "Wake metrics: : :" is part of the mentioned*
*future work and not relevant for wake center location analyzed in this paper?*
30 *Then they could greyed-out or removed.*
>>The Gaussian fit was greyed out. As indicated, wake metrics include the wake centre location.

Page 13, line 7-8: The velocity deficit is introduced with two variables (delta U and
v_d).
35 >>This has been corrected

*Page 13, line 11: What does the "X D +/- 20 m" in the brackets mean?*
>> This was changed in response to the comments of a previous reviewer.

40 *Page 13, line 13-14: The sentence "Then the plane of radial velocities is discretized*
*into 20 m horizontal planes and a mean radial velocity is computed." needs some*
*clarifications: a) Are the radial velocities averaged or a corrected velocity depending*
*on az/el of the beams and wind direction? b) If I understood correctly, the 20 m are*
*referring to the vertical height of each plane – so that in the end a U_0 is gained that*
45 *only depends on y and z?*
>>Yes. That is correct. A background radial wind speed is determined for each distance downstream at discrete heights of 20 m. This has been clarified p14, l16 and we also refer to Figure 6.
'At each of these downstream distances (± 20 m) the radial wind speeds at each x location (lateral displacement distance from a direct transect to the wind turbine) and z height (where z is defined from the elevation of the wind turbine hub-height) are retrieved
50 for each 10-minute period. Then the vertical plane of radial velocities is discretized into 20 m horizontal planes and a mean radial velocity is computed for each 20m plane (see Figure 6).'

*Page 13, line 15-16: The detection of the wake center should be explained in more detail. As I understood it, the algorithm searches*

*the nearest local maximum from x=z=0.I did not understand what is meant with "refining that location by checking it moving*

55 *around the grid cells". From the next paragraph, I understand that an interpolation takes place before detection of the wake center*

*- that should be mentioned here.*

>>This has been clarified p14, l19:

'Anomalies from that 'background' profile are then interpolated using cubic spline interpolation and used in the wake centreline identification. The location of the wake centre is determined using the maximum velocity deficit anomaly for each height starting its search at the expected location (*WTHH*). The location is refined by moving from that location horizontally replacing the wake
5   centre if the new grid cell velocity deficit is greater than the previous maximum velocity deficit value. Once locations have been checked in each 20 m horizontal plane, the algorithm moves to the next vertical plane and checks that searching for the maximum velocity deficit value. The algorithm assumes the wake has moved further downstream than the immediate rotor plane after the double bell wake shape is expanded into a near Gaussian shape (Barthelmie et al., 2003) and hence that there is a well-defined centre. '

10   *Fig. 5: It should be stated that axis are distance from the lidar.*

>>This is in the caption to Figure 5. 'Each panel is 1000 m by 1000 m with the Galion located at (0,0) and depicts line of sight (radial) wind speeds from each elevation angle; 7, 9, 10.5, 12, 13.5, 15, 16.5, 18, 19.5, 21 and 23°.'

*Fig. 6 and Fig. 7: x and z could be used for axis labels.*

15   >> The current version that states explicitly 'Height from WTHH' and 'Lateral distance' is clearer

*Page 16, line 13: comma before but*

>> A comma is not needed here. Commas are not required before the word 'but' unless two independent clauses are joined by the conjunction

*Section 4.2 (headline): I believe "Data availability" would better describe the content of*
*this section.*

Changed to '4.2 Data processing methodology and data availability'

25   *Page 22, line 4: I do not understand what is meant with "wake centre of gravity".*

This was changed to p23, l3' B) type where the wake centre is split but the wake centre is broadly identified by the algorithm,'

*Page 22, line 12: "at" instead of "as".*

>> Done

*Page 22, line 14: comma before but*

>> Commas are not required before the word 'but' unless two independent clauses are joined by the conjunction

*Page 22, line 14: I believe it should be either "identifies a part of the wake" or "identifies*
35   *parts of the wake".*

>> Done.

*Page 26, line 11: Remove inner brackets and replace with "or" or "; also see" (I first*
*thought it is a reference to section 1.2 in Whiteman and Doran).*

40   >>Done

*Fig. 12: The example of wake type B (and comparing it with with type D) makes me concerned about misdetections. How does the wake center of type B cases develop downwind – i.e does it evolve in a continuous manner or is the wake center "jumping around" for successive downwind distances? For an automated wake detection, a*
5 *quality flag system providing indication on the reliability of the detected wakes would be a great enhancement (beside the above, other possible routes could be the spatial standard deviation compared with the amplitude of the detected wake center or the number of local minima/maxima above a certain threshold).*

>> There is a lot of work to do on these type B wakes. Thanks for the interesting suggestions.

10 Given the length of the current manuscript we decided to focus on metrics of the A type cases here and follow with a detailed investigation of the much more complex B cases in a subsequent manuscript. For your interest, the lobes seem to be coherent as they move downstream but a much more comprehensive analysis is needed to confirm this and the other detail of their characteristics and behavior.

15 *Fig. 16, lower panel: The usage of the symbols to show the hour of day is not working, because they cannot be read within the figure. And I could not make sense of the text in the top right (I got the z/L intervals, but what is the meaning of the S, N and U?). In my opinion, the time dependency is better illustrated in Fig. 15 and therefore Fig. 16 should focus on the dependency with the stability parameter. One idea could be, to*
20 *pick one exemplary downwind distance and plot distance from WTHH vs. z/L. Then a linear fit /correlation (if significant) could quantify the relation with stability parameter. The remaining downwind distances could be reported in the text or in a table.*

>> The main point is to show both the general tendency that is really clear in the above figure and the variability by both hour and stability class. This helps to show that while the averages are really distinct there is still variability. It is not important to be able

25 to distinguish the numbers on each line. We've added a detailed description of the two legends and also rewritten this section to illustrate the important points made by this Figure more clearly as below. The changes are clear in the tracked version of the paper provided:

'Thus, there is a tendency for the wake centre to be higher in stable than unstable conditions (Figure 16). The daytime hours of 09:00 to 17:00 UTC are largely associated with unstable conditions. During these hours of the day, the mean wake centre is slightly

30 above the equivalent *WTHH* (+3.5 m) at a downstream distance of 2 *D* to an average of -50 m by a distance of 4.5 *D* (Figures 15 and 16). In stable conditions that prevail during 18:00 to 07:00 UTC, the mean wake centre is an average of +13.5 m from *WTHH* at 2 *D* and -33 m at 4.5 *D*. Most stable hours have wake centre trajectories that are higher than the majority of those in unstable hours (see the groupings of stable wake centre trajectories in blue colours vs the unstable wake centre trajectories in red colours in Figure 16). Despite this clear signal, there is also variability both in the grouping of individual hours into different stability classes

35 and the height of the wake centre trajectory by stability. For the most extreme case of lofting, in the hour 21.00 (marked L) conditions are stable and the average of all values for the wake centre is +27 m at 2 *D* downstream but then descends to -38 m by 4.5 *D*. In contrast, at 8:00, an hour that is defined as near-neutral (marked 8), and at 13:00, an hour that is very unstable (marked D), the wake centre drops below -40 m after 3 *D* downstream. The behaviour of wakes is clearly very complex but despite a large amount of scatter, there is a consistent relationship between the value of *z/L* and the wake centre height for each downstream

40 distance with wake centres in stable conditions being higher (Figures 15 and 16). '

*Fig. 17: The color coding is unfortunate, because two similar greens and purples are*
*used. Grey scales or a linear color map (e.g. blue -> red) would better illustrate a*
*dependency. Similar to the comment on Fig. 16, the dependency could be quantified*
*with a linear fit and correlation (if significant).*

5  >>The point is to show the general tendency of the relationship with wind speed. We evaluated various options and came up with the use of distinct symbols which, when combined with the colours, has improved this figure.

*Section 5 (headline): "Summary" would be a better description for the content of the*
*section.*

10  Changed to 'Summary and Conclusions'

*Page 28, line 10: What is physical forcing?*
Physical forcing is generally understood to mean forces affecting on flow like the pressure gradient force, Coriolis force acting rather than thermally-generated mechanisms. To clarify this sentence we have changed it to 'There are very few near-neutral

15  conditions and the lack of a clear diurnal signal in wind speed is indicative of multiple scale impacts on the flow, including both physical and thermal forcing. '

*Page 28, line 18: Insert "to" between "applied identify"*
>> Done

[revised manuscript text omitted]

A previous study analysed a more limited sample of data from scanning Doppler lidars operated from the northeast ridge during from thea 2015 campaign at the Perdigão site and sought to examine the behaviour of wakes from the wind turbine (Menke et al., 2018). While our statistical assessment of 1120 wake cases from 2017 show that the mean wake height is lower as it moves downwind from the crest in unstable conditions than in stable conditions, that previous study of 21 cases suggest that in unstable conditions the wakes are elevated, that they propagate horizontally in near-neutral or stable conditions and can be downwards if gravity waves are present (Menke et al., 2018). There are several differences in the research methodologies that can account for this apparent discrepancy. These are summarized in Table S1 and include:

1) The studies are not for the same period with the same wind speed conditions. We have shown that in virtually all cases with an inflow wind speed < 12ms$^{-1}$, the wake centre is initially lofted (i.e. at a height greater than the wind turbine hub-height centreline) and that by about 3 $D$ downstream most wakes start to descend. In low wind speeds (as shown for the unstable case in wakes are initially lofted and remain above the horizontal line of the wind turbine hub-height further downstream. Of the 21 cases presented in (Menke et al., 2018) 14 showed upwards movement while 6 were propagated horizontally and 2 downwards (no information was provided regarding at which distances downstream).

2) The location and configuration of the Doppler lidars is fundamentally different. The earlier study analysed data from lidars scanning in the near horizontal (i.e. at considerable height above the intervening valley and ridge slopes). In this paper data are reported from a scanning Doppler lidar configured to scan upslope towards the wind turbine. Thus, the ability to track the wind turbine wakes differs. Use of the scanning lidar in the diamond pattern allows a horizontal view of the wake for about 4-5$D$ but does not permit wake tracking if it move substantially from the horizontal plane. Further, the flow field detected and the transect scans showreflects a combination of the wake with the larger scale flow such as the hill wake which propagates over much longer distances than a wind turbine wake. Conversely, the lidar configuration used in this work while use ofrelies principally on arc scans as positive displacements from the horizontal and permits tracking for a wider variety of wake centre locations but does not permit horizontal tracking of the wake beyond about 4.5 $D$.

3) There are major differences in the stability classification from the WRF model at 3 km resolution and in that from in situ sonic anemometer measurements.

Table S1. Comparison of measurements and results from this study and those in (Menke et al., 2018)

| | This study | Menke et al. 2018 |
|---|---|---|
| Stability measurements | $z/L$ from hub-height sonic anemometer at ridge top | Stability defined using 3 km resolution simulations with the Weather Research and Forecasting mode. HWRF, height not given |
| Stability definition | stable $L$<1000 m, neutral $L$±|1000 m|, unstable $L$>-1000 m | stable $1/L$ > 0.01, neutral 0.01 > $1/L$ > -0.01, unstable $1/L$< -0.01 |
| Data period | 27 Jan – 30 Jun 2017 | 8 May- 26 Jun 2015 |
| 10-minute periods analyzed | 1120 | 21 |
| Scan type | Aarc scans at multiple elevation angless | Nnear-horizontal diamond & transect scans |
| Distance from WT over which wakes are tracked | 2-4.5 $D$ | 1-~30 $D$ |

---

## Author Response (AR2)

Response to Associate Editor Decision: Publish subject to minor revisions (review by editor)

(28 May 2019) by Jose Laginha Palma
Comments to the Author:

All questions raised by the referees were answered and the manuscript may proceed for publication.

At his stage, I have two suggestions.

Suggestion 1:
Concerning the "Supplemental information", it should be embedded in the manuscript final version, as a new subsection or in subsections 1.3 or 1.4?

We have moved the discussion from Supplementary Material into section 4.4 so that it is included with the discussion of the results.

Suggestion 2:
The authors should improve the flowcharts in pages 24 and 25. For instance: (1) flowcharts are not clear and do not convey the sequence of operations as they should; and (2) symbols (triangles, parallelograms, circles, etc) do not follow the standards and decision branches (Yes and No) are not always identified. Input/output operations, process and logical decisions, etc should be represented by the appropriate shape.

The Figure has been extensively modified (Figure 4).
Please see the tracked changes below.

[revised manuscript text omitted]